# Geographic Distribution Pattern Determines Soil Microbial Community Assembly Process in *Acanthopanax senticosus* Rhizosphere Soil

**DOI:** 10.3390/microorganisms12122506

**Published:** 2024-12-04

**Authors:** Mingyu Wang, Xiangyu Xing, Youjia Zhang, Xin Sui, Chunying Zheng

**Affiliations:** Engineering Research Center of Agricultural Microbiology Technology, Ministry of Education & Heilongjiang Provincial Key Laboratory of Ecological Restoration and Resource Utilization for Cold Region & Heilongjiang Provincial Key Laboratory of Plant Genetic Engineering and Biological Fermentation Engineering for Cold Region & Key Laboratory of Microbiology, College of Heilongjiang Province & School of Life Sciences, Heilongjiang University, Harbin 150080, China; wmy022234@163.com (M.W.); 13325152141@163.com (X.X.); zdzhidian@outlook.com (Y.Z.)

**Keywords:** *Acanthopanax senticosus*, rhizosphere microbiome, geographic factors, environmental variables

## Abstract

The geographic distribution patterns of soil microbial communities associated with cultivated *Acanthopanax senticosus* plants in Northeast China were investigated. High-throughput sequencing revealed that the diversity and community assembly of bacterial and fungal communities in the inter-root soil varied significantly with geographic location. The study found that bacterial communities were predominantly assembled through stochastic processes at most sites, while fungal communities showed greater variation, with both stochastic and deterministic processes involved. The complexity of bacterial–fungal co-occurrence networks also varied with longitude and latitude, demonstrating both positive and negative interactions. PICRUSt 2.0 and FUNGuild were used to predict the potential functions of soil bacterial and fungal microbiota, respectively, during different land use patterns. The average taxonomic distinctness (AVD) index indicated varying degrees of community stability across sites. Key microbial taxa contributing to community variability were identified through Random Forest modeling, with *Bacteriap25* and *Sutterellaceae* standing out among bacteria, and *Archaeorhizomyces* and *Clavaria* among fungi. Soil chemical properties, including pH, TN, TP, EC, and SOC, significantly correlated with microbial diversity, composition, and co-occurrence networks. Structural equation modeling revealed that geographic distribution patterns directly and indirectly influenced soil chemical properties and microbial communities. Overall, the study provides insights into the geographic distribution patterns of soil microbial communities associated with *A. senticosus* and highlights the need for further research into the underlying mechanisms shaping these patterns.

## 1. Introduction

Geographic distribution patterns not only reveal the spatial characteristics of biodiversity, but also reflect the complex relationships between ecological environments and biological communities [1,2]. Geographic distributions constitute a unique research horizon that allows a deep exploration of the dynamic evolutionary patterns of biomes, ecological adaptation mechanisms, and complex interactions among species [3]. With the intensification of global climate change and human activities, it is becoming increasingly important to assess changes in geographic distribution patterns of soil microorganisms, as these represent key components of ecosystems [4]. Soil microorganisms play an indispensable role in maintaining soil health, promoting plant growth, and regulating ecosystem functions, and are vital life forces in the Earth’s biosphere [5,6]. A deeper understanding of their geographical distribution and the factors influencing this can assist with better predictions of potential distribution areas and can provide a scientific basis for soil microbial biodiversity conservation.

Studies on microbial biogeography have demonstrated that geographic factors, such as latitudinal gradients and soil chemistry, play a pivotal role in shaping microbial community composition and diversity [7,8]. For example, deterministic processes like niche differentiation and species sorting often interact with stochastic factors, influencing the distribution of microbial taxa across latitude [9]. However, drivers remain poorly understood in ecosystems associated with medicinal plants, where rhizosphere microorganisms are integral to plant health and productivity. Investigating microbial community assembly in the rhizosphere of different medicinal plants can uncover how plant-microbe interactions evolve under distinct ecological pressures. While recognizing the close relationship between ecosystems and biological communities as reflected in geographical distribution patterns, the central position of soil microorganisms in this complex network should not be overlooked. Based on the neutral theory, the formation and development of soil microbial communities is a dynamic and multifaceted process, which involves complex mechanisms including the selective colonization of species, ecological niche differentiation and adaptive responses to the environment [10,11]. Microorganisms settle in specific ecological niches according to their own characteristics and environmental needs, and this non-random process is influenced by multiple factors [12]. The adaptability of species determines whether they can successfully colonize and reproduce in a given environment, where they need to adjust their physiological and ecological strategies in response to environmental pressures [13]. These mechanisms are interrelated and work together to shape complex and diverse biomes [14]. Some soil microorganisms form symbiotic relationships with plants in the inter-root microenvironment, where they maintain ecosystem stability and prosperity, profoundly influencing nutrient uptake and enabling plant growth; as such, these organisms are critical for plant adaptation to environmental changes and defense against diseases [15]. For example, Zuo et al. (2023) pointed out that close interactions between soil microorganisms and plants enables both to jointly cope with complex changes in the external environment [16]. Systematic studies of these interactions can further reveal the synergistic evolution of soil microbes and plants.

In traditional Chinese medicine, *Acanthopanax senticosus* has high medicinal value with anti-fatigue, anti-inflammatory, and cardiovascular activity [17]. It has been widely cultivated in the past decades and its growth is stimulated by inter-root soil microorganisms [18]. Studies on similar plants, such as *Panax ginseng* and *Glycyrrhiza uralensis*, have demonstrated that rhizosphere microorganisms significantly influence plant nutrient uptake and stress tolerance. Soil microorganisms form a close symbiotic relationship with the root system of these plants, building a complex ecosystem that exerts a profound influence on the plant’s growth [19]. This interaction between microorganisms and *A. senticosus* form a complex and subtle ecosystem, in which the microorganisms not only promote nutrient uptake and growth but may also play a key role in the formation and accumulation of medicinal components. However, few studies have explored the geographic distribution patterns and community assembly process of rhizosphere microorganisms in *A. senticosus*. Understanding these dynamics can provide insights into the ecological adaptations of *A. senticosus* and its interactions with soil microbes.

To this extent, we analyzed the inter-root soil microorganisms of *A. senticosus* cultivated at different sites of Northeast China. An objective of this study was to test the suitability of the Illumina Miseq platform for detecting community assemblage within different types of rhizosphere soils. We hypothesized that (1) the relative importance of community assembly between rhizosphere bacteria and fungi varies due to different sensitivity to soil chemistry; and (2) rhizosphere communities less influenced by species sorting exhibit higher species coexistence and community stability. The study not only revealed complex interactions between soil microorganisms and *A. senticosus*, but also provides important clues to further explore how these interactions affect plant growth, nutrient uptake, and stress tolerance. In addition, the study increases our understanding and ability to respond to potential impacts of environmental changes on soil ecosystems and agricultural production of *A. senticosus*.

## 2. Materials and Methods

### 2.1. Study Sites

Soil samples for this study were taken from six different sites in Heilongjiang Province, northeastern China, as shown in Figure 1. These sites were located at Stone River Forestry, Nancha County, Yichun City (YC); Shuguang Forestry, Fangzheng County (FZ) and Harbin Yuquan Hunting Ground, Acheng District (AC), both near Harbin City; Qingmeishan Forestry, Mishan City (MS) near Jixi City; and Qing’an County Forestry (QA) and Mingshui County Forestry (MIS), both near Suihua City. In April 2023, sampling was conducted on the inter-root soils of agriculturally grown *A. senticosus*. Using a specially designed sterile soil sampler, 10 soil samples of 15 × 15 cm were collected at each site. The samples were immediately transported to the laboratory where they were divided into two portions. One portion was stored at −80 °C for the subsequent determination of the soil microbial community. The other portion was naturally air-dried at ambient temperature and passed through a 2 mm sieve prior to determination of soil chemical properties.

### 2.2. Determination of Soil Chemical Properties

The soil pH was determined by means of a pH meter (Sartorius PT-21, Shanghai, China), employing a soil-to-ddH_2_O ratio of 1:2.5 [20]. The total nitrogen (TN) content of the soil was measured with a Vario Max CNS elemental analyzer (Elementar Analysensysteme GmbH, Hanau, Germany) [21]. The determination of total potassium (TK) and total phosphorus (TP) was carried out using an Inductively Coupled Plasma Emission Spectrometer (Agilent 5100 ICP-OES, New York, NY, USA) [21]. Soil organic carbon (SOC) was analyzed through oxidation with potassium dichromate using an external heating program [22]. Soil electrical conductivity (EC) was measured by a conductivity meter (Multiparameter SevenExcellence™, Tianjin, China) [23]. Soil cation exchange capacity (CEC) was measured after adding ammonium acetate (NH_4_OAc) at pH 7 [24].

### 2.3. Determination of Plant Biomass and Secondary Metabolites

Growth measurements of *A. senticosus*: We measured root length (cm), Stem length (cm), and plant Height (cm) using a tape measure. The thickness of current year branches (cm) and stems (cm) were measured using vernier calipers. Also, the number of leaves was counted and Leaf length (cm) and leaf width (cm) were measured using a tape measure, which led to the calculation of Leaf area (cm^2^). In addition, we recorded the number of branches as well as the length (cm) of the current year’s branches, and used these indicators as a basis for evaluating the growth of *A. senticosus* [25]. The field spectrometer-based retrieval [26] used the PROSPECT radiative transfer model [27] to estimate Chlorophyll content.

The filtered plant extracts were analyzed using HPLC (Waters Corp., Milford, MA, USA), with 50 mM NaH_2_PO_4_ (spiked with 8 mM octane sulfonic acid sodium salt) (Solvent A) and methanol (100%) (Solvent B) as the mobile phase, to quantify Chlorogenic acid, Quercetin, Hyperoside, and Syringin. The solvent A reaction was maintained at a pH of 3 by adjusting with H_3_PO_4_ [28]. For column separation, the solvents were equilibrated at an A:B ratio of 80:20. Detection of these compounds was carried out using ultraviolet light at a wavelength of 254 nm over a runtime of 30 min. Detector sensitivity was calibrated to 0.005 absorbance units full scale. The analysis was conducted using a C-18 column (Symmetry, Waters Corp., Milford, MA, USA) at a flow rate of 1 mL min^−1^, and the concentrations of the target metabolites were quantified based on peak areas [29]. Prepare a stock solution of 5 mg mL^−1^ of each standard with methanol and store at −20 °C. Calibration curves were generated by preparing five concentrations (0.01, 0.025, 0.20, 0.50, and 1.50 mg mL^−1^) of all chemicals through a step-wise dilution. As part of the performance assessment for HPLC, seven replicates of the liquid working standards were analyzed to estimate method detection limits (MDLs) and relative standard errors. The MDL values for the target compounds were calculated based on the absolute mass contained in a 1 mL standard. The concentrations of the target compounds in plant samples were subsequently converted to mg kg^−1^.

### 2.4. DNA Extraction and High-Throughput Sequencing

Total DNA was extracted from the soil with the Omega Soil DNA Kit (M5635–02, Omega Bio-Tek, Norcross, GA, USA). This DNA was used to amplify the bacterial 16S rRNA region using primers 338F (5′-ACTCCTACGGGAGGCAGCAG) and 806R (5′-GGACTACHVGGGTWTCTAAT) [30,31]. The fungal ITS rRNA region was amplified using primers ITS1 (5′-CTTGGTCATTTAGAGGAAGTAA) and ITS2 (5′-GCTGCGTTCATCGATGC) [32]. Following purification of the amplicons by standard procedures, they were sequenced on a MiSeq-PE300 platform (Illumina, San Diego, CA, USA), employing the Illumina MiSeq Reagent Kit v3 at Shanghai Personal Biotechnology Co., Ltd. (Shanghai, China) [33]. The reads were analyzed with QIIME 2 [34]. First, raw reads were demultiplexed using the demux plugin and then primer sequences were trimmed with the cut adapt plugin. Dada2, with default settings, was employed to filter and denoise the sequences, employing learnErrors, derepFastq, dada, and mergePairs to ensure sequence quality and mitigate sequencing errors. Singleton amplicon sequence variants (ASVs) were discarded. Bacterial taxonomic identities were identified utilizing the Silva v132 database (http://www.arb-silva.de) [35] and fungal taxonomic identities were identified utilizing the GlobalFungi database (https://globalfungi.com/) [36]. We used PICRUSt 2.0 and FUNGuild to predict the potential of the function of soil bacterial and fungal microbiota during different land use patterns, respectively [37,38].

### 2.5. Bioinformatic and Statistical Analysis

Geographic location mapping was performed using ArcGIS v10.8 [39]. In this process, we first imported datasets containing geolocation data, which may include coordinate information such as longitude and latitude. Then, in ArcGIS, we used this coordinate information to create geospatial layers to accurately position each point in the dataset on the map. In order to present the data more intuitively, we also visualized the data points, such as setting different colors, sizes, or shapes to represent different data attributes or classifications. A one-way analysis of variance (ANOVA) was used to compare differences in soil chemical properties and alpha-diversity of soil microbial communities at different latitudes and longitudes [40]. Mapping metacoder evolutionary trees was performed with the help of “metacoder”, “ggclusterNet”, “Tidyverse” and “phyloseq” packages in the R software (version 4.4.1) [41]. We constructed a Random Forest classification modeling with the help of “randomForest”, “psych”, “reshape2” and “ggplot2” packages in the R software (version 4.4.1) [42]. We first implement the random forest algorithm through the “randomForest” package, which is popular for its excellent classification and regression performance. Before modeling, we use the “psych” package to conduct a comprehensive descriptive statistical analysis and visualization of the data, in order to grasp the overall characteristics and distribution patterns of the data. Meanwhile, the “reshape2” package helps us easily reshape and organize the data to ensure that the data format meets the modeling requirements. Finally, in the model training and evaluation phase, we use the “ggplot2” package to create intuitive graphs to clearly show the model performance evaluation results. To quantify co-occurrence networks, the “igraph” package was employed. For visualization, Gephi (http://gephi.github.io/) was used to generate network maps [43]. We simulated random networks with identical node and edge counts as our empirical network, and analyzed topological indices through 999 iterations [44]. This approach allowed us to assess whether specific network properties deviated significantly from randomness. Furthermore, subnetworks within the global co-occurrence network were identified and partitioned based on predefined node sets, such as representative nodes for each sample, utilizing the subgraph function provided by the “igraph” package in the R software (version 4.4.1) [45]. We used “Hmisc”, “minpack.lm”, “stats4”, and “grid” packages in the R software (version 4.4.1) to construct neutral community model (NCM) [11]. The Sloan Neutral Community Model, which was created using the “vegan” package in the R software (version 4.4.1) aided in clarifying the processes involved in the formation of soil bacterial and fungal communities [46]. Specifically, the “Hmisc” package provides us with rich data processing and statistical analysis functions, which help us better prepare and preprocess the data, and lay a solid foundation for subsequent modeling [47]. The “minpack.lm” package is a tool for nonlinear least squares fitting, which supports fitting with bounding constraints, allowing us to handle various complex situations more flexibly in the model construction process. Key parameters in the model, such as metacommunity size (N) and migration rate (m), are set according to the study objectives and data characteristics. These parameters will be used to characterize stochastic processes and species dynamics in the community. The contribution of neutral processes to community assembly (e.g., R^2^ values) will be calculated by fitting the observed distribution of species abundance to the distribution predicted by neutral theory. Meanwhile, the “stats4” package also plays an important role in model construction by providing a rich set of functions and tools to support statistical inference such as maximum likelihood estimation, thus helping us to estimate model parameters more accurately. Finally, the “grid” package provides us with powerful graphical drawing functions, which enable us to visualize the model results and conduct further data visualization and analysis [46]. The βNTI values were calculated with the help of “iCAMP”, “ggpubr”, “NST”, “picante” and “ape” packages in the R software (version 4.4.1) [48]. The βNTI values were determined by contrasting the actual community structure with the predicted theoretical structure based on a null model. If the βNTI value falls within the range of −2 to 2, stochastic processes prevail; when the value of βNTI is less than −2 or greater than 2, deterministic processes are prevalent. Under the established constraints, the community data are randomized by reallocating species to different samples or adjusting the interspecies relationships while preserving certain ecological characteristics, such as species richness. To obtain a stable null distribution, the randomization process is typically repeated multiple times (e.g., 1000 times or more) [49]. After each randomization, corresponding community structure indices (e.g., βNTI, RCbray, etc.) are calculated [50]. By constructing the community assembly null model, ecologists can compare the observed community structure indices with the null distribution to identify any discrepancies.

Structural equation modeling (SEM) incorporating categorical variables was established utilizing the “lavaan” package in the R software (version 4.4.1) [51]. Firstly, it was ensured that the collected data met the requirements of the study, and multivariate normal distribution test and multiple covariance test were performed to satisfy the assumptions underlying the SEM analysis and to ensure the accuracy of the model parameter estimation [52]. Subsequently, model identification and setup were performed to determine the unique solutions for the free parameters of the model and to set up the model structure based on the path diagram in the statistical software. Model parameters were estimated using methods such as maximum likelihood estimation (ML) or generalized least squares (GLS). The fit of the model to the data was assessed by a goodness-of-fit metric, and model diagnostics were performed to check the reasonableness of the estimates. If the model is poorly fitted, corrections are made based on the diagnostic results, such as adjusting variables or changing the model structure, and re-estimated and re-assessed until a satisfactory fit is achieved [53,54]. Nonlinear fitting models of latitudinal and longitudinal changes and soil microbial diversity, βNTI, and AVD indices were constructed using “ggplot2” and “dplyr” packages in the R software (version 4.4.1) [55,56]. The “dplyr” package was used to clean the data, deal with missing values and outliers, and perform data type conversion and data transformation. Then, we selected a suitable nonlinear model according to the research purpose, fitted it with “nls” or “glm” function, and optimized the model parameters. The effectiveness of the model was evaluated by a goodness-of-fit metric, and the residual distributions and diagnostic plots were examined. Finally, scatter plots were drawn using the “ggplot2” package and nonlinear fitting curves were superimposed to present the results of the study. AVD indices for soil bacterial and fungal communities were calculated using “vegan”, “anosim”, and “vegdist” packages in the R software (version 4.4.1) [57]. The functions provided by the “vegan” package are used to analyze community data, which may include information on species richness, diversity, and community structure. The ‘anosim’ function is used to perform a non-parametric similarity analysis (ANOSIM) to test whether there are significant differences in community structure between groups. Finally, the “vegdist” function is used to calculate the distance or similarity between communities, which is one of the key steps in calculating the AVD index. Stacked plots were generated utilizing the “ggplot2” package in the R software (version 4.4.1) [58]. Principal Coordinate Analysis (PCoA), a multivariate statistical technique used to visualize and interpret complex datasets, was conducted by employing the “vegan” package in the R software (version 4.4.1) [59]. Correlation heat maps were drawn using “Psych”, “pheatmap”, “ggplot2”, “ggcorrplot”, “corrplot”, and “corrgram” packages in the R software (version 4.4.1) [60]. RDA was performed using the “vegan”, “ggrepel”, and “ggplot2” packages in the R software (version 4.4.1) [61]. Multivariate statistical analyses were performed through the “vegan” package. The “rda” function in the “vegan” package is the core tool for performing redundancy analyses and constructing RDA models by specifying response variables and explanatory variables. The “summary” function allows the extraction of the results of the RDA. Finally, the “ggplot2” package is used in conjunction with the RDA results extracted by the “vegan” package to produce an esthetically pleasing and informative RDA ordination plot.

## 3. Results

### 3.1. Soil Chemical Properties, Plant Biomass and Secondary Metabolites

The soil of the inter-root regions of cultivated *A. senticosus* plants collected at various locations in Northeast China varied considerably for pH, TN, TP, SOC and CEC (Table 1). The pH was highest at the MIS site and lowest at the YC site, while levels of TN, TP and SOC were highest in soil collected from the YC site and lowest in the MIS soil. CEC was highest at the MS site and lowest at the FZ site (Table 1). The growth characteristics of *A. senticosus* varies from site to site. Root length, Stem length, Height, Annual branch thickness, Stem thickness, and Chlorophyll content, as well as Annual branch length of *A. senticosus* were the highest at the YC site, whereas the highest number of leaves and number of branches were reached at the FZ site. In addition, Leaf length, leaf width, and Leaf area of *A. senticosus* showed the highest at MS site (Appendix A). There were significant differences in the root secretions of *A. senticosus* at different sites, with Chlorogenic acid being highest at site FZ and AC being lowest. While Hyperoside and Syringin were highest in QA site (Appendix A).

### 3.2. Changes of Soil Microbial Diversity and Composition in Different Geographic Distributions of Acanthopanax senticosus Rhizosphere Soils

The alpha-diversity of the bacterial and fungal communities present in the inter-root soil samples differed significantly among sites. In particular, the Simpson index of soil bacterial communities was lower in QA soil compared to the other samples (Appendix A; Table 2). The Shannon and Simpson indices of the soil bacterial community were lower in QA compared to the other sites, indicating different levels of diversity in the soil. The Richness index of soil fungal communities was high in MS, FZ, and AC compared to the others, and the Chao index was particularly low in MS and QA, indicating that fungal species richness and community complexity also differed between the samples (Appendix A).

The average taxonomic distinctness (AVD) was calculated for the detected bacterial and fungal biomes, to reflect the species diversity and uniqueness of the species composition of the communities (Figure 2). The soil collected from the MIS site exhibited the highest bacterial AVD index, with that of the MS site being the lowest (Figure 2a). Among the soil fungal communities, the YC site resulted in the highest AVD index, with the QA site giving the lowest AVD index (Figure 2b). Principal coordinate analysis (PCoA) was used to evaluate the similarity in composition of the soil bacterial and fungal communities (Figure 3; Appendix A). The distance of datapoints in the graph relates to the differences between the communities. This indicated that the identified bacterial community in the soil from the MIS site was quite different from that of the other five sites, while the bacterial communities from the MS, FZ, and YC sites shared a certain similarity (Figure 3a). For the fungal communities, the fungal assemblages present at QA and YC were quite distinct from each other and from the remaining four sites. However, a certain degree of similarity was observed in the soil fungal communities populating the MS and FZ sites (Figure 3b).

The obtained sequence reads were attributed to taxonomic entities. Among the soil bacterial communities, the order of *Chthonomonadales*_sp. was enriched in MIS (green), *Candidatus_Magasanikbacteria* was enriched in FZ (green), Bdellovibrionaceae was enriched in MS (green), Caulobacterales was enriched in AC (green) (Appendix A). Sphingomonadaceae was enriched in YC (green) and Saccharomycetes was enriched in QA (brown). Among soil fungi, Taphrinomycetes was enriched in MIS (green), Mortierellomycetes was enriched in FZ (green), Cladophialophora was enriched in MS (green), Trichomeriaceae was enriched in AC (green), Hyalodendriella was enriched in YC (green). Ploettnerulaceae was enriched in QA (brown) (Appendix A).

At the genus level, in nearly all bacterial communities *Candidatus_Udaeobacter* was dominant, with a relative abundance of 41–73%, except for MIS, where its fraction was 22%. The second most abundant genus was *RB41*, which was found at 10–20% in MS, FZ, and AC, but at 36% in MIS and only 5% to 6% in QA and YC. The community of MIS was also enriched for *Sphingomonas*, while *Bryobacter* and *Candidatus Solibacter* were enriched in FZ and YC. Lastly, *Bacteroides* reached a relative abundance of 18% in YC only (Figure 4a) Of the fungal genera, *Russula* was strongly dominant in AC soil, reaching a relative abundance of 64% compared to 7–23% in the other sites. *Mortierella* was dominant in MIS (54%) and reached 25–35% in the other soils, except for QA where its fraction was only 8%. *Archaeorhizomyces* showed a remarkably higher abundance in FZ (34%) compared to all other sites (Figure 4).

Random Forest classification modeling was employed to identify key bacterial and fungal taxa that play pivotal roles in shaping the soil biomes (Figure 4c,d). this identified that in the soil bacterial communities, the classes *Bacteriap25* and *Acidimicrobiia* and the family *Sutterellaceae* were the key taxa responsible for the significant differences in soil bacterial communities at the six different sites (Figure 4c). Among the soil fungal communities, the genera *Archaeorhizomyces, Clavaria* and *Harmajaea* were identified as key contributors explaining the differences between the soil fungal communities at the six different sites (Figure 4d).

### 3.3. Soil Microbial Co-Occurrence Networks

Microbial co-occurrence networks were performed at the phylum level, for bacteria and fungi (Appendix A). The bacterial co-occurrence networks were quite complex for four of the six soil communities (Appendix A). The soil samples of AC and QA resulted in much simpler bacterial networks than the others. The maximum number of bacterial nodes (281) and edges (35) was identified in soil from YC. In all networks, the bacterial interactions were mostly positive (81–93%), in YC only positive interactions were identified (Appendix A; Appendix A). The phylum *Acidobacteriota* ranked number 1 in four of the six soil samples (*Proteobacteria* scored highest in MS and *Verrucomicrobiota* ranked number 1 in AC). Except for the relatively simple AC and QA networks, the combination of *Proteobacteria* and *Acidobacteriota* represented between 51% and 54% of the nodes in the other soils.

The fungal networks (Appendix A) were much simpler than those representing bacterial co-occurrences, containing between 6 and 43 nodes, and between 3 and 22 edges. The simplest fungal network was found for UC and the most complex for FZ. Only FZ and MIS soil fungal co-occurrence networks contained both positive and negative relationships, while the remaining sites only reported positive network relationships. In all fungal co-occurrence networks, Ascomyeota was ranked first. The bacterial fungal mutualistic co-occurrence network relationships were also assessed (Figure 5). Now, the network was the most complex for AC soil, with 179 nodes and 2113 edges. The networks of MS and MIS were of similar complexity, while the networks of FZ, QA, and YC were the simplest. Thus, the bacterial–fungal mutualistic relationships were strongest in the soil from AC, and weakest in FZ (Figure 5).

### 3.4. The Community Assembly of Soil Microbial Communities

The Between-community Nearest Taxon Index (βNTI) was used to assess ecological processes during community assembly. This revealed different dominant mechanisms at different sites for the bacterial community assembly process (Figure 6). Based on a value of βNTI between −2 and +2, it can be concluded that in most locations the community assembly of inter-root soil bacterial community was the result of a stochastic process. Only at the MIS location was this process clearly governed by deterministic processes, while the bacterial community of MS was at the border of stochastic and deterministic (Figure 6a). The assembly patterns of the fungal communities showed greater variation between sites. The assembly in MIS soil was dominated by stochastic processes, and in MS, FZ and AC soil the assembly process was mainly controlled by deterministic processes; borderline findings were reported for QA and YC (Figure 6b).

Appendix A demonstrated the process of bacterial and fungal community assembly in the inter-root soil of *A. senticosus* by Neutral Community Model (NCM). This identified that the bacterial community assembly process in the soil of YC is about to be transformed from a stochastic to a deterministic process, whereas the bacterial and fungal community assembly processes at the other sites are dominated by stochastic processes (Appendix A).

### 3.5. Factors Influencing Changes in Soil Microbial Communities

A correlation heatmap was constructed to visualize correlations between top ranking bacterial and fungal genera and soil chemical properties (Figure 7). Of the bacterial genera, *RB41*, *Sphingomonas*, *Haliangium*, *Ellin6055*, *Terrimonas* and *Dongia* showed significant positive correlations with soil pH, while *Candidatus_Solibacter* and *Bryobacter* correlated negatively (Figure 7a). Seven identified correlations between fungal genera and pH were negative: *Tomentella*, *Lnocybe*, *Cortinarius*, *Podila*, *Pseudosperma*, *Cenococcum* and *Thelephora*, with a positive correlation identified for *Sclerotinia*, *Waitea* and *Ceratobasidium* Figure 7b. Strong positive correlations were observed between *Rhodoplanes* with TN, TP, EC, and SOC, while *Ceratobasidium* had a significant negative correlation with TN, TP, EC, and SOC (Figure 7).

The geographic latitude and longitude of the sampled sites was used to fit into a model to assess their effect on the bacterial and fungal communities. The results showed that the species richness and diversity of the bacterial and fungal communities varied significantly along latitudinal and longitudinal gradients in Northeast China (Appendix A). The model predicted a nonlinear fit of geographic location, with different trends for bacterial and fungal Choa1 indices; differences in the effect of longitude and latitude were also identified. Similar analyses were performed for the Shannon index (Appendix A), AVD index (Appendix A) and the βNTI index (Appendix A). With the increase in latitude and longitude, the AVD index of soil bacteria and fungi first showed an increasing trend, reaching a peak to then gradually decrease (Appendix A). The βNTI index of soil bacteria and the Chao1 index of soil fungi produced a decreasing and then increasing trend (Appendix A). This pattern of increase followed by a decrease reveals that the diversity and variability of soil bacterial communities have complex patterns in response to different geographic locations.

A Mantel test identified a strong correlation between the bacterial and fungal composition of the soil samples with latitude and longitude (Figure 8). Further analysis showed that the AVD index of soil bacteria present in the inter-root zone of *A. senticosus* strongly correlated with soil pH and TP, whereas the AVD index of soil fungi highly correlated with EC and TN. In addition, it was identified that the bacterial βNTI index significantly related to EC, while the fungal βNTI index strongly correlated with pH and TP. Finally, the co-occurrence networks of both soil bacteria and fungi were significantly correlated with TN, TP, EC, and SOC (Figure 8).

Significant correlations were found between the diversity and richness of soil bacterial communities and Syringin as well as Stem thickness. In addition, the richness of soil bacterial communities showed significant correlations with Chlorophyll content (Appendix A). Soil bacterial community βNTI was significantly correlated with Chlorogenic acid, Syringin, Height, Annual branch thickness, Stem thickness and Annual branch length (Appendix A). Soil fungal community diversity was significantly correlated with Hyperoside, Stem length, Chlorophyll content and Leaf area. Soil fungal community richness was significantly correlated with Hyperoside and Leaf length. Soil fungal community βNTI was significantly correlated with Stem length and Annual branch thickness (Appendix A). RDA analyse showed that soil pH has a significant impact on the soil bacterial community at the MIS site, while SOC, EC, and TN exert a substantial influence on the soil fungal community at the same location (Appendix A; Appendix A).

Finally, fitting all the data into a Structural equation modeling (SEM), which revealed that geographical distribution patterns exert both direct and indirect impacts on soil chemical properties and soil microbial communities in the inter-root zone of *A. senticosus* (Figure 9). Additionally, alterations in soil chemical properties were associated with distinct geographical distribution patterns, and these can further modulate soil microbial diversity, ultimately shaping the community assembly process of soil bacterial and fungal communities (Figure 9).

### 3.6. Prediction and Analysis of Soil Microbial Community Function

We used PICRUSt2 to predict soil bacterial function. In the study, it was found that aromatie_compound_degradation, aromatie_hydrocarbon_degradation, hydroearbon degradation, aerobie_chemoheterotrophy and Chitinolysis were significantly enriched at the FZ site. Meanwhile, the cellulolysis function showed enrichment only at the QA site. These results suggest that bacterial communities in different sites and treatment groups are characterized by unique functional enrichment. anoxygenic_photoautotrophy, anoxygenic_photoautotrophy_S_oxidizing, nitrous_oxide_denitrification, nitrite_denitrification and denitrification were enriched in MIS site (Figure 10a). We used FUNGuild to predict soil fungal function. Among the YC sites were those fungi capable of functioning as Animal Pathogen–Endophyte–Fungal Parasite–Plant Pathogen, Ectomycorrhizal–Fungal Parasite–Plant Saprotroph and Plant Pathogen–Undefined Saprotroph. Dung Saprotroph–Soil Saprotroph–Wood Saprotrop, Bryophyte Parasite–Dung Saprotroph–Ectomycorrhizal–Fungal Parasite and Plant Saprotroph–Wood Saprotroph were enriched in AC site. Soil Saprotroph–Undefined Saprotroph, Wood Saprotroph, Endophyte–Plant Pathogen–Wood Saprotroph, Plant Pathogen–Wood Saprotroph and Endophyte–Epiphyte–Fungal Parasite–Insect Parasite were enriched in QA site (Figure 10b).

PcoA analysis of the functional predictions of soil bacterial and fungal communities showed that the functional predictions of soil bacterial communities did not differ significantly across the six different sites; however, there was a significant difference in the functional predictions of soil fungal communities between the QA site and the other sites (Figure 3). Redundancy analysis (RDA) showed that the differences in soil bacterial and fungal community function predictions were more pronounced across geographic distribution patterns influenced by plant biomass. The RDA model indicating that plant biomass metrics (Leaf area, Branch number, Chlerophyll content and Number of leaves) varying along the RDA1 axis were the dominant factors contributing to the variability in bacterial community functional prediction (Appendix A). In addition, the RDA1 axis explained 10.1% of the total variance in the prediction of soil fungal function, indicating that plant biomass metrics varying along the RDA1 axis (Leaf area, Branch number and Number of leaves) were the dominant factors contributing to the variability in the prediction of fungal community function. chlophyll content, Annual branch thickness, Stem length, and number of leaves were the dominant factors contributing to the variability in the prediction of fungal community function. Annual branch thickness, Stem length, Height, Stem thickness, and Root length had more pronounced effects on soil fungal community function prediction at the MS site (Appendix A).

## 4. Discussion

### 4.1. Soil Microbial Diversity and Composition

The Simpson index of the fungal communities present in the rhizosphere of *A. senticosus* was the lowest in location QA (Appendix A). Larger values of the Simpson index indicate a lower diversity, i.e., a higher concentration of species, while smaller values indicate a higher diversity with a more uniform distribution of species [62]. This indicated that soil fungal diversity was highest and evenly distributed in QA. This may be due to the high latitude as well as low longitude of the QA site. This location creates specific climatic and ecological conditions that may not be conducive to the survival of a wide range of fungi, which in turn affects microbial community structure and diversity in the soil [63,64]. As a result, the detected soil fungal communities at the QA site have a relatively even distribution of species, lacking dominant species. When environmental conditions are favorable for a wide range of fungi, the ecology is relatively stable and lacks the dramatic environmental changes that drive fungal community diversification [65,66]. This stability leads to a dynamic equilibrium in the fungal communities that keeps the Simpson’s index low, but this equilibrium may be disturbed if environmental conditions change, thus affecting the Simpson index.

The AVD Index is a key indicator describing the stability of the soil microbial community within the root zone of *A. senticosus*, and its stability is essential for assessing ecosystem health and predicting the effects of environmental change on microbial communities [67]. A higher AVD index usually implies poorer community stability and results in increased sensitivity to environmental changes, which may be accompanied by reduced microbial diversity and impaired ecological functions; in the opposite situation, a lower AVD index indicates strong community stability and adaptability to environmental changes, as well as a rich microbial diversity with relatively well-developed ecological functions [68,69]. The results showed that soil bacterial communities in MS plots had the lowest AVD indices, followed by QA and YC (Figure 2). Again, this may be related to the high latitude and longitude location of QA and YC, and the specific climatic and environmental conditions, such as temperature and humidity that apply here. This may have indirectly contributed to the accumulation of SOC, whereas higher SOC implies that the soil is fertile and rich in organic matter, which provides an adequate food source for microorganisms [70]. In such an environment, microbial communities may exhibit higher stability and thus lower community variability as abundant food resources reduce competitive pressure among species [71]. The MS site is located at a low latitude combined with a high longitude. Measurements of soil chemistry indicate that the MS site has a low pH. Acidic soils may limit the growth of certain bacteria, which could lead to an increase in the dynamics of the microbial community, where some species are unable to adapt, while others adapt to acidic environments and dominate [72].

From the PCoA analysis, we observed that the soil bacterial community at the root interface of *A. senticosus* at the MIS site was clearly different from that of the other sites (Figure 3a), which was likely related to the lower TN, TP, and SOC content at the MIS site (Table 1). Specifically, the lack of these nutrients may have limited bacterial growth and colonization, leading to changes in bacterial community structure and diversity. In order to adapt to this nutrient-limited environment, bacteria may have adopted special survival strategies, such as changing resource utilization, adjusting metabolic pathways, or forming symbiotic relationships [73,74]. At the same time, the bacterial community may have undergone adaptive selection in the face of the pressure of a low-nutrient environment, which led to the formation of a unique bacterial community [75,76]. In addition, the low-nutrient environment exacerbates interspecific competition among bacteria, prompting them to form closer cooperative relationships to jointly utilize resources. Ultimately, the special environmental conditions at the MIS site exerted strong selection pressure on the bacterial community, shaping the unique bacterial community structure at the site.

In addition, based on the PCoA analysis, there was a significant difference between the soil fungal communities at sites QA and YC, and the composition of the soil fungal communities at these two sites differed from that of the other sites (Figure 3b). This may be due to the fact that the longitude of sites QA and YC differed significantly from the other sites. QA and YC are in the middle of the longitude range. At the same time, soil types at different sites vary depending on geographical location, which further affects the composition of soil fungal communities [77]. Therefore, at the sites with significant longitude differences between QA and YC, soil fungal communities may have undergone ecological niche differentiation and evolved different survival strategies to adapt to their respective environments [78]. In addition, this difference in longitude may also lead to restricted species migration and dispersal, allowing specific species of fungi to survive in a particular range of longitude thus leading to differences in the composition of soil microbial communities.

In the soil bacterial community, *Candidatus_Udaeobacter* ranked high in soil bacterial communities in all geographical locations (Figure 4a). As a widespread and highly ranked genus in a wide range of environments, it possesses a wide range of metabolic pathways and is able to utilize a variety of carbon and energy sources, which allows it to find suitable survival strategies in a wide range of ecosystems. Secondly, *Candidatus_Udaeobacter* has gained an advantage over other microorganisms through strategies such as rapid growth, efficient resource utilization or production of metabolites that inhibit the growth of other microorganisms [79], further consolidating its position in the community. In addition, *Candidatus_Udaeobacter* showed better tolerance to acidic pH, and thus was able to survive better in acidic soils in the Northeast [80]. Notably, *Candidatus_Udaeobacter* was able to maintain high relative abundance under a variety of climatic conditions, which further proved its strong climatic adaptability [79]. However, when the geographic location was at MIS, the abundance of *Candidatus_Udaeobacter* dropped to the second place and was replaced by *RB41*, which became the top-ranked genus (Figure 4a). SOC and TN, as the core indicators of soil fertility, play a crucial role in microbial survival and reproduction [81]. The MIS site, however, had the lowest SOC content and relatively low TN content. This low nutrient environment may limit the growth of certain microorganisms, including *Candidatus_Udaeobacter*. These nutrients are indispensable elements for microorganisms to carry out their life activities, and their lack can hinder their growth. Under nutrient-limited conditions, competition among microorganisms is intensified. *RB41* may be able to distinguish itself from *Candidatus_Udaeobacter* by dominating in nutrient-poor environments due to its greater ability to acquire and utilize nutrients [82]. *RB41* may also be able to compete with *Candidatus_Udaeobacter* in a nutrient-poor environment due to its greater ability to acquire and utilize nutrients.

Among the fungal communities, *Russula* ranked first in abundance at the AC site, while *Mortierella* ranked first in abundance at the MIS site (Figure 4b). *Russula* may be better adapted to the soil environment at the AC site, benefiting from its superior adaptation to the specific soil conditions at the site and thus dominating the site. Specifically, it may be highly compatible with soil physico-chemical properties (e.g., pH, TN, TP) at the AC site, while forming favorable symbiotic relationships with other fungi and being more competitive [83]. In contrast, *Mortierella* may be more adapted to the soil environment of the MIS site. Particularly considering that the MIS site is relatively nutrient-poor with low SOC and TN content [84], *Mortierella* may exhibit enhanced nutrient acquisition and utilization by virtue of its unique physiological mechanisms and metabolic pathways, allowing it to survive and colonize this nutrient-restricted environment [85]. Fungal communities are dynamic systems in a constant process of succession. In this process, *Russula* may be more adaptable in the specific environment of an AC site and therefore be able to maintain a dominant position in the community succession at that site [86]. *Mortierella*, on the other hand, may have shown greater adaptability in the low-nutrient environments of the MIS site, an adaptation that allowed it to emerge as the dominant genus in the intense competition at that site. This adaptive difference is one of the important reasons for the different abundance rankings of *Russula* and *Mortierella* at different sites, and it reflects the response and adaptive strategies of fungal communities to environmental changes.

Random forest classification modeling indicated that *Sutterellaceae* and *Bacteriap25* were the key bacterial taxa contributing to the variability in the inter-root soil bacterial community of *A. senticosus* at the six investigated sites (Figure 4c). *Sutterellaceae* members have excellent ecological niche adaptation capacity, and are well adapted to the special conditions of the inter-root soil of *A. senticosus*, such as nutrients, humidity and pH, which explains they are able to proliferate here [87]. Meanwhile, as inter-root bacteria, *Sutterellaceae* can establish positive interactions with the root system of *A. senticosus*, both assisting the plant in nutrient uptake and promoting its growth, as well as obtaining necessary energy and nutrients from root secretions. In addition, within the microbial community, these bacteria may regulate the growth of other microorganisms by producing antimicrobial substances or competing for nutrients, thus occupying an important position in the community, and may also form symbiotic relationships with other microorganisms to jointly maintain the stability of the community. *Bacteriap25* is able to efficiently utilize specific nutrients or organic matter in the inter-root soil of *A. senticosus* due to its unique metabolic properties, which makes it stand out in the competition for resources [88]. Meanwhile, specific factors in the soil environment, such as a suitable temperature, humidity, and soil type, may provide favorable conditions for *Bacteriap25* to dominate the inter-root soil of *A. senticosus*. The interaction with the root system of the plants not only provides *Bacteriap25* with necessary nutrients but may also assist the plant to enhance stress tolerance and promote nutrient uptake, creating a mutually beneficial relationship.

In addition, the fungi *Archaeorhizomyces* and *Clavaria* were the key genera shaping the soil fungal community (Figure 4d). We speculate that this may be due to the ability of their mycelium to penetrate deep into the soil and build an intricate network structure. This physical interpenetration not only enhances the aeration of the soil, but also increases its water retention capacity, thus effectively improving soil structure [89]. This improvement is extremely important for plant growth and the health of the entire soil ecosystem. At the same time, these two fungal genera play an integral role in the carbon cycle: they are able to break down organic matter, releasing carbon into the soil, which may subsequently be absorbed and utilized by plants or enter the atmosphere through respiration [89,90]. Thus, the genera *Archaeorhizomyces* and *Clavaria* may occupy a key position in the global carbon cycle and climate regulation [90]. In addition, their presence and activity assist to maintain the diversity of soil microbial communities, which in turn enhances ecosystem stability. In short, these two fungal genera are essential for maintaining the balance and stability of the entire ecosystem.

### 4.2. Soil Microbial Co-Occurrence Network and Community Assembly

The bacterial and fungal co-occurrence network interactions surrounding the roots of *A. senticosus* was most complex at the AC location (Figure 5). This site is located at a lower latitude and (except for MIS) at a lower longitude, which allows the coexistence of a high diversity of bacterial and fungal species. This diversity not only results in a richness of potential interactions, but also increases the complexity of the identified networks [91]. These different species of microorganisms, each with unique metabolic pathways and ecological functions, weave an intricate network of interactions among them [92]. At the same time, these microorganisms play a pivotal role in soil nutrient cycling and energy flow [93]. Especially in AC soils, these processes are exceptionally active, further promoting close collaborations among microorganisms. For example, fungi are able to decompose organic matter, releasing nutrients that can be utilized by bacteria, while bacteria may provide fungi with needed nutrients through processes such as nitrogen fixation. Together, such complementarities and collaborations sustain the prosperity and stability of microbial communities in soil. All of the above processes have the potential to promote increased complexity of soil bacterial and fungal interactions.

The complexity of bacterial and fungal co-occurrence networks surrounding the roots of *A. senticosus* at the AC site can be attributed to the unique ecological environment that promotes niche differentiation and competition among diverse microbial species. Influenced by its geographical location (lower latitude and lower longitude), the soil environment likely harbors a more diversified array of microhabitats [94], providing ample living space and resources for different bacterial and fungal species. However, the limited availability of resources fuels intense competition among microorganisms, each striving to occupy the most favorable niche for itself [95]. In this competitive landscape, various microbial species develop unique metabolic pathways and ecological functions to reduce direct competition with others, thereby achieving niche differentiation [96]. Concurrently, niche competition drives the dynamic equilibrium of microbial communities, where certain species may dominate due to their strong adaptability and efficient resource utilization [97], while others may adjust their ecological strategies or seek new niches to cope with competitive pressures. This dynamic balance not only maintains the diversity of microbial communities but also equips them with the ability to respond to environmental changes.

The complexity of the co-occurrence network describing bacterial interactions was only highest in YC soil (Appendix A). This was probably mainly due to its unique soil environment and geographic conditions: the relatively low soil pH provided a favorable growth environment for acidophilic bacteria, while high levels of TN and TP provided abundant nutrients [76,98], contributing to the prosperity and diversification of the community. Meanwhile, high EC values contributed to the electrochemical exchange among bacteria, while high SOC provided more energy for the bacteria. In addition, the specific climatic and ecological conditions brought about by the high latitude of the YC region create favorable conditions for the formation of bacterial symbiotic networks [99,100]. Together, these factors provide suitable growth conditions and nutrient sources for bacteria, thus promoting the diversity of bacterial communities and the complexity of symbiotic networks [101]. The most complex symbiotic network of soil fungi was found in the FZ soil (Appendix A). The FZ region, at a high longitude and low latitude, allowed fungi to successfully adapt to specific environmental conditions through ingenious symbiotic strategies, which enable them to utilize resources more efficiently in resource-limited environments through symbiotic relationships [102]. This tight integration of environmental adaptation and symbiotic strategies not only enables fungi to stabilize their coexistence in variable environments, but also greatly improves the stability and complexity of symbiotic networks. Ascomyeota was ranked first in the soil fungal co-occurrence network at all sites (Appendix A). This is due to their multifaceted strengths; their extreme ecological adaptability allows Ascomycota to survive and dominate in a variety of soil environments [78]. These fungi have excellent nutrient acquisition and decomposition capabilities, effectively breaking down organic matter and providing nutrients to the soil ecosystem [103], thus playing a key role as a decomposer. Furthermore, the influence of Ascomycota in the soil is further enhanced by their ability to form symbiotic or parasitic relationships with other organisms [78], such as combining with plant roots. Their strong competitive ability also helps them to stand out from the fierce fungal competition and solidify their primacy in the soil fungal symbiotic network.

It is noteworthy that the co-occurrence networks of soil bacteria and fungi in the YC region exclusively comprise positive relationships (Appendix A), a unique phenomenon closely linked to niche differentiation. The specific latitude and longitude of the region, along with its distinctive climatic conditions, soil characteristics, and biogeographic patterns [104,105], not only provide abundant nutrients and ideal growth environments for microorganisms but also facilitate niche differentiation among them. The low pH and high levels of total nitrogen (TN), total phosphorus (TP), electrical conductivity (EC), and soil organic carbon (SOC) in YC soils enable different types of microorganisms to select and occupy the most suitable niches based on their environmental adaptability, thereby reducing interspecies competition and promoting community stability and harmony [6]. Furthermore, the unique biogeographic pattern of high latitude and longitude regions offers a relatively stable ecological environment conducive to the coexistence of specific bacteria and fungi, and niche differentiation further minimizes direct competition among them, thus maintaining this positive relationship. Niche differentiation not only enhances microorganisms’ resource utilization efficiency but also strengthens community stability and diversity, serving as one of the crucial factors contributing to the positive correlations observed in the soil microbial co-occurrence networks of the YC region.

The bacterial community assembly in the rhizosphere of *A. senticosus* at the MIS site was dominated by deterministic processes, while in FZ, AC, QA, and YC soil it was dominated by stochastic processes (Figure 6). The MIS region has a high latitude but a lower longitude compared to the other sites. The fact that bacterial communities in this area are assembled in a deterministic manner is mainly due to the specific effects of the local soil characteristics. Soil types in these regions have unique physicochemical properties, such as less acidic pH, low TN and TP levels, and low CE and SOC levels (Table 1), which together with specific mineral compositions selectively promote the growth of certain bacterial populations while limiting the development of others [106,107]. In addition, the specific distribution of nutrients in the soil provides growth advantages for specific bacterial populations, further cementing the certainty of the community structure. At the same time, environmental factors such as suitable temperature and humidity conditions, as well as specific concentrations of oxygen and carbon dioxide in the soil [108], provide favorable environments for the survival of certain bacterial populations, which together contribute to the deterministic assemblage of the bacterial community [72]. The predominance of deterministic processes in the specific regions of FZ, AC, QA, and YC may be attributed to the fact that the similar latitude and longitude maintains a dynamic equilibrium of the soil microbial populations at these sites, where the competitive and cooperative relationships among the various microorganisms are constantly changing. In such a changing environment, random events, such as mutations and migrations, have the potential to profoundly affect the evolution of a microbial community. At the same time, soil bacterial communities exhibit a high degree of genetic diversity, implying the presence of numerous different genotypes and phenotypes of bacteria. This rich diversity further amplifies the importance of stochastic processes in community evolution, as each bacterial species is unique in the way it adapts and responds to its environment. It also establishes the dominance of stochastic processes at these sites.

We found that for the MIS site, the soil fungal community assembly processes differed significantly from those in the other five sites (Figure 6; Appendix A). stochastic processes predominated in the MIS area, which may be related to the specific environmental conditions associated with the location of the site. Since fungal spores have a strong dispersal ability and can be widely spread by a variety of means, including wind, water, and animals, spores may be more likely to be randomly distributed under geographic conditions such as those at MIS, resulting in a community structure that is dominated by stochasticity [109,110]. In addition, soil fungi in the MIS area may not yet have developed strong competitive or symbiotic relationships with each other, which increases the importance of stochastic factors in community formation [111]. From another perspective, soil fungal communities in the MIS area may be in an early stage of succession, when the community structure has not yet stabilized and various fungal species are still adapting and adjusting to each other [112]. At this stage, stochastic processes have an important influence on the initial composition and subsequent development of the community [113].

### 4.3. Key Regulators Influencing Changes in Soil Microbial Communities

Soil pH, a crucial factor influencing microbial growth, dictates the adaptability of different bacteria. Among the identified members of soil bacterial communities, some genera, exemplified by *Terrimonas* and *Sphingomonas* (Figure 7), exhibit significant positive correlations with soil pH. *Terrimonas* is adapted to higher pH environments and demonstrates enhanced metabolic activity and growth rates within specific pH ranges [114]. *Sphingomonas*, on the other hand, engages in complex interactions with other microbial communities in specific pH environments, encompassing symbiosis and competition, which collectively facilitate its survival [115]. Notably, *Sphingomonas* may collaborate with nitrogen-fixing bacteria that convert atmospheric nitrogen into usable ammonia, providing a valuable nitrogen source for *Sphingomonas* [116]. In return, *Sphingomonas* potentially secretes growth factors, vitamins, or other beneficial substances that strengthen this mutualistic relationship. In terms of competition, *Sphingomonas*’ exceptional adaptability to specific pH environments enables it to more efficiently utilize soil resources [117]. This efficient resource utilization not only grants it an advantage in resource competition but also may further inhibit the growth of other competing microorganisms by producing antimicrobial compounds with enhanced activity under specific pH conditions. This competitive advantage ensures *Sphingomonas’* dominance in soil microbial communities, potentially explaining its higher relative abundance in MIS samples with higher soil pH compared to others (Figure 4a, Table 1). Niche competition, as a significant driver of changes in microbial community structure, not only shapes the success of *Sphingomonas* but also influences the stability and diversity of the entire soil ecosystem.

In the soil fungal community, the strong negative correlation observed between *Tomentella* and *Pseudosperma* with soil pH (Figure 7) highlights an intriguing facet of niche competition. *Tomentella* produces soil enzymes that exhibit higher activity in acidic environments, which not only promotes its metabolism and growth but also enhances the efficiency of related metabolic pathways. Additionally, acidic conditions may increase the availability of certain nutrients, further aiding the survival and proliferation of *Tomentella*. This adaptation to acidic soils likely explains their higher relative abundance in YC soils, which have a relatively lower pH. Similarly, the strong negative correlation between *Pseudosperma* and soil pH can be attributed to the impact of acidic environments on membrane stability, with *Pseudosperma* demonstrating adaptability by maintaining the stability and function of its cell membranes, allowing for normal metabolic activities, ion balance, and preserving the integrity of ion channels and transporters. Conversely, environments with lower acidity or alkalinity can induce conformational changes in membrane lipids and proteins, disrupting membrane fluidity and permeability. Given *Pseudosperma*’s unsuitability for non-acidic environments, this accounts for its high relative abundance in AC soils. These findings emphasize the importance of niche competition in shaping the composition of soil fungal communities, with *Tomentella* and *Pseudosperma* securing ecological niches through adaptation to specific pH conditions, providing them with a competitive advantage over other fungi less suited to these environments. This ecological differentiation not only promotes their own survival and growth but also contributes to the overall stability and diversity of the soil ecosystem.

The effect of soil chemical properties is demonstrated by the bacterial genus *Rhodoplanes*, which are strongly positive correlated with TN, TP, and SOC, while fungal *Ceratobasidium* negatively correlated with these levels (Figure 7). It is quite possible that *Rhodoplanes* are able to efficiently break down and utilize organic matter in the soil, converting these complex organic molecules into valuable nutrients, as *Rhodoplanes* are more adapted to nutrient-rich environments [118], so they are more active in soils with high TN, TP, and SOC contents. When present, *Ceratobasidium* disrupt soil aggregates, leading to soil structural deterioration, thereby exposing and accelerating the decomposition of otherwise protected SOC [119,120]. This would reduce the SOC content and may release greenhouse gasses, such as carbon dioxide. At the same time, its decomposing activity toward organic matter alters the physical and chemical properties of the soil, including soil texture, porosity, and pH, which in turn affects soil aeration, water movement, and nutrient element forms and availability. Although *Ceratobasidium* activities release elements such as nitrogen and phosphorus [121], its efficient utilization of these elements leads to a decrease in the amount of TN and TP available in the soil for plants and other microorganisms, which ultimately has a significant impact on the content and availability of TN, TP, and SOC in the soil, highlighting the important role of *Ceratobasidium* in the soil ecosystem.

The results presented here show that the soil bacterial and fungal community alpha-diversity and community assembly processes in the investigated locations were strongly responsive to geographic distribution patterns. The Chao1 index of soil fungi strongly decreased and then increased with increasing longitude (Appendix A). Sites with a higher longitude may be accompanied by changes in soil pH and organic matter content, which have profound effects on the adaptation and food sources of soil fungi [122,123]. Soil pH is an important factor affecting fungal growth, and different fungi have different adaptations to pH [66,124]. With the gradual change in soil pH from more acidic to less acidic, the fungal populations may shift from members adapted to lower pH environments to those adapted to higher pH [125,126,127]. Such differences not only affect the number of fungi but may also lead to an adjustment in the structure of the fungal community [65]. In addition, gradual changes in organic matter content also had a significant effect on fungi [128]. Organic matter is the main food source of soil fungi, and changes in its content are directly related to the growth rate and reproductive capacity of fungi [129]. Depending on the longitude, if the soil organic matter content decreases, the fungi may be limited in growth due to food shortage, which in turn leads to a decrease in their diversity. Conversely, if the organic matter content increases, fungi will proliferate, potentially contributing to their increased diversity.

The limited availability of nutrients, water, and organic matter in soil leads to the formation of unique ecological niches among different microbial species, which vary in their efficiency and modes of resource utilization. As environmental conditions such as soil pH and organic matter content change with longitude, the ecological niches of soil fungi and bacteria differentiate accordingly [83]. According to the competitive exclusion principle in niche theory, microorganisms with similar niches compete fiercely for limited resources, potentially leading to the elimination or reduction in some species. To avoid such competitive exclusion, soil microorganisms further differentiate their niches to reduce competitive pressure [97]. The variation in the Chao1 index of soil fungi with longitude reflects this shift in fungal diversity driven by niche differentiation and competition [130]. In resource-rich environments, multiple fungal species may coexist and compete with each other, whereas in resource-poor environments, fungal species diversity decreases, and competitive pressure diminishes [131]. The greater the overlap in niches among fungi, the more intense the competition; conversely, niche differentiation alleviates competition, favoring the coexistence of multiple fungal species [132]. Thus, changes in soil fungal diversity are the joint outcome of soil environmental heterogeneity, microbial niche differentiation, and competitive exclusion.

In addition, the soil fungal community βNTI index showed a nonlinear trend with increasing latitude, as evidenced by an increase followed by a decrease and then an increasing trend again (Appendix A). To explain this identified trend, two perspectives come to mind: environmental gradient and community dynamics. Different latitudes are accompanied by environmental differences including temperature and humidity, which can directly affect the growth and reproduction of fungi [133]. At the same time, the fungal community undergoes succession over time, and the succession stages and ecological niche adjustments may be different at different latitudes, which together affect the βNTI index and leads to the observed nonlinear trend. This complex pattern of change reflects the diversity and dynamics of assembly processes of soil fungal communities in different latitudinal environments.

Using the Mantle test, we identified a strong correlation between the composition of inter-root soil bacteria and fungi of *A. senticosus* and latitude and longitude (Figure 8). The internal mechanism of this correlation includes multiple factors: changes in latitude and longitude bring about a diversity of climatic conditions and soil types, which affect the microbial growth environment [134]. In addition, the geographical location affects biogeochemical cycles, resulting in differences in nutrient cycling and organic matter decomposition rates, which further affect the microbial food chain and survival environment [135]. Meanwhile, the competitive symbiosis and community succession among microorganisms are also affected by latitude and longitude, and these factors are intertwined to create the community structure of soil microorganisms. The key environmental factors, total nitrogen (TN), total phosphorus (TP), electrical conductivity (EC), and soil organic carbon (SOC), all strongly correlated with the symbiotic network of soil bacteria and fungi within the root zone of *A. senticosus* (Figure 8). The contents of TN and TP, as essential nutrients, directly determine the nutrient sources of microbial communities, which in turn affect the microbial diversity and activity, and enables the symbiotic network to support a greater number of species of microorganisms [136,137]. Meanwhile, SOC, as a carbon and energy source for microorganisms, is directly related to microbial metabolism and reproduction, thus regulating the species and number of microorganisms in the symbiotic network. These factors work together to maintain the stability and diversity of the microbial symbiotic network.

### 4.4. Prediction and Analysis of Soil Nicrobial Community Function

The FZ site was significantly enriched in aromatic compound degradation, aromatic hydrocarbon degradation, hydrocarbon degradation, oxidative heterotrophy and chitinolysis, which are closely related to the decomposition and metabolism of organic matter (Figure 10a). It was hypothesized that the soil at the FZ site was rich in aromatic compounds, hydrocarbons, and other organic substances, which provided abundant carbon and energy sources for specific degrading bacteria, and contributed to the significant enrichment of these bacteria [138]. Meanwhile, the soil at this site was acidic (pH 5.53 ± 0.09) and had a high electrical conductivity (EC 16.8 ± 0.395 dS/m), which indicated that the soil was high in salinity, and these physicochemical characteristics jointly acted on the microbial community, further strengthening the enrichment of bacteria with specific degradation functions at the FZ site. QA sites are significantly enriched for cellulolytic functions, thanks to the fact that soils may be enriched in plant residues and cellulosic material, providing an ideal growth environment for cellulose-breaking bacteria. These bacteria are particularly active under specific soil conditions, promoting efficient cellulose decomposition, which is essential for carbon cycling and maintenance of soil fertility [139]. The MIS site was significantly enriched for non-oxygenic photosynthetic autotrophs, non-oxygenic photosynthetic autotrophs (sulfur oxidation), nitrous oxide denitrification, nitrite denitrification, and denitrification, which are usually closely associated with the sulfur and nitrogen cycles. It is hypothesized that the soils at the MIS site may have specific redox conditions that provide suitable growth environments for the bacteria that carry out these processes [140,141]. For example, sulfur oxidation and denitrification processes typically occur in anoxic or anaerobic environments, which may be related to the unique soil physicochemical properties of MIS sites. In addition to pH and EC, soils at MIS sites may have specific physicochemical properties such as SOC, TN, TP, and TK, which together act on the microbial community to promote significant enrichment of specific functional bacteria.

Soil fungal communities at the YC site exhibit a high degree of diversity and complexity, encompassing a wide range of functional types from animal pathogens, endophytic fungi, fungal parasites, to plant pathogens, as well as exophytic mycorrhizal fungi, plant saprophytes, and undefined saprophytes (Figure 10b). This wide distribution of ecological roles emphasizes the uniqueness and richness of the soil environment at the YC site. In terms of physicochemical properties, the soil at the YC site has some remarkable characteristics: it has a low pH (5.31 ± 0.05), an acidic environment that is usually favorable for the growth of acid-loving fungi [142,143]; at the same time, the TN and TK contents of the soil are at a high level, which provides an adequate source of nutrients for the fungi and promotes their growth and metabolism [144]; in addition, organic carbon serves as an important carbon source for fungal growth—its high content provided a better survival environment for soil fungi [145]. In addition, the high cation exchange capacity of the soil indicated that the soil was capable of retaining water and fertilizer, which was further conducive to the survival and reproduction of the fungal community. The fungal functions enriched at the AC site mainly included fecal saprophytes, soil saprophytes, wood saprophytes, as well as moss parasites, ectomycorrhizal fungi, fungal parasites, and plant saprophytes, which were mainly associated with the saprophytic process, suggesting that the soil fungal community at this site was dominated by saprophytes. The high SOC content of the AC site is an important carbon source for fungal growth, and saprophytic fungi mainly obtain energy and nutrients through the decomposition of organic matter; therefore, the high SOC content may provide sufficient sources for saprophytic fungi to promote their growth and reproduction [146]. Considering the soil physico-chemical properties of the YC and AC sites, especially the organic carbon content and pH, we found that these conditions had a significant effect on the composition and function of the fungal community. This suggests that subtle changes in the soil environment may lead to significant shifts in the structure of fungal communities, which in turn affect their ecological functions. It is noteworthy that despite differences in some soil physicochemical properties, the fungal communities of some sites showed a strong preference for saprophytic life. This suggests that saprophytic processes may be a key driver of fungal community structure and function in similar soil environments. Therefore, the influence of saprophytic processes should be fully considered when predicting and managing soil fungal communities.

Significant correlations were found between the diversity and richness of soil bacterial communities and Syringin as well as Stem thickness. In addition, the richness of soil bacterial communities showed significant correlations with Chlorophyll content (Appendix A). The mechanistic mechanism of these correlations may involve interactions between plants and soil microorganisms. Plants release a variety of compounds, including Syringin, to the soil through root secretions, which act as carbon sources and signaling molecules that can influence the composition and activity of soil microbial communities [18,147]. Meanwhile, Stem thickness and Chlorophyll content, as indicators of plant growth and health status, may indirectly reflect plant effects on soil microbial communities [148,149]. Specifically, plants with larger Stem thickness may have stronger root systems, providing more habitats and nutrient sources for soil microorganisms, thus promoting the diversity and richness of bacterial communities [18,150]. Plants with high Chlorophyll content, on the other hand, may have higher photosynthetic efficiency and nutrient uptake capacity, releasing more organic matter into the soil and further promoting soil bacterial community richness [151,152,153]. Therefore, these significant correlations between plant traits and soil bacterial communities may be the result of plant–soil microbial interactions and mutual influences.

Moreover, Soil fungal community diversity was significantly correlated with Hyperoside, Stem length, Chlorophyll content, and Leaf area. Soil fungal community richness was significantly correlated with Hyperoside and Leaf length (Appendix A). This may stem from the fact that plants release a variety of compounds, including Hyperoside, into the soil through root secretions and residue decomposition, which act as carbon sources and signaling molecules that have a significant impact on the structure and composition of soil fungal communities [18,153]. Fungal communities utilize these compounds for growth and metabolism, resulting in patterns of fungal community diversity and richness associated with specific plant traits [154,155]. Specifically, plant morphological traits such as Stem length and Leaf area may reflect plant growth status and resource acquisition capacity, which in turn influence the diversity and richness of soil fungal communities [149,156]. Meanwhile, Chlorophyll content, as an indicator of plant photosynthesis capacity, may also indirectly affect the composition of soil fungal communities by influencing nutrient uptake and utilization by plants [3,157]. Therefore, the significant correlation between soil fungal community diversity and abundance and plant traits may be the result of interaction and co-evolution between plants and soil fungi.

PcoA analysis showed significant differences in predicted soil fungal community function between the QA site and the other five sites (Appendix A). This was mainly attributed to the unique soil physico-chemical properties of site QA: the lowest TK content, which may limit fungal enzyme activity and growth, and thus affect community composition and function [158]; and the highest EC, indicating high salt content, which may promote the growth of salt-tolerant fungi and affect community diversity and function [159]. Meanwhile, QA sites had low pH, TN, and CEC, which may promote the growth of acid-tolerant fungi but limit fungal species that rely on high nitrogen and carbon content. These combined factors worked together to result in significant differences in soil fungal community function at the QA sites. RDA showed that plant biomass metrics (Leaf area, Branch number, and Number of leaves) were the dominant factors contributing to the variability in bacterial and fungal community functional prediction (Appendix A). The increase in Leaf area means that the area available to the plant for photosynthesis increases, which in turn enhances the productivity and metabolic activity of the plant [160]. This change leads to the release of more organic matter into the soil by the plant, such as root secretions and detached leaf fragments, which provide a rich source of carbon and energy for soil microorganisms [161]. Thus, the increase in Leaf area may have promoted the diversity and activity of soil microbial communities, which in turn has implications for the prediction of their function. Bacteria and fungi obtain carbon sources and energy by decomposing organic matter such as plant residues and root secretions, whereas the increase in plant biomass provides more food sources for microorganisms and promotes their growth and reproduction [162]. Different species of microorganisms have different preferences and utilization efficiencies for specific types of organic matter, which leads to the diversity of microbial community structures and differences in function. In resource-rich environments, competition between microorganisms may be exacerbated, but cooperation between them may also be promoted. Certain bacteria may dominate by producing antibiotics that inhibit the growth of other microorganisms, while other microorganisms may increase the efficiency of their utilization of organic matter by forming symbiotic relationships [163]. Such competitive and cooperative relationships further influence the prediction of microbial community function.

However, the relatively narrow latitudinal gradient of this study is a limitation, as it may restrict a comprehensive understanding of microbial community responses to environmental changes at different latitudes. Large-scale studies with broader latitudinal ranges could provide more comprehensive insights into the geographic distribution patterns and adaptive mechanisms of microbial communities. Future research should expand the latitudinal gradient to validate and complement these findings, offering a more comprehensive view of how microbial communities adapt to different environmental conditions across latitudes. This will further enhance our understanding of microbial community geography and provide valuable data for ecological and microbiological studies.

## 5. Conclusions

This study provides new insights into the geographic distribution patterns of soil microbial communities in the rhizosphere of *A. senticosus* plants in Northeast China, based on large scale field sampling, high-throughput sequencing, and multiple statistical analysis. Our result showed that the diversity and community assembly of the inter-root soil microbial communities showed significant nonlinear trends with increasing longitude and latitude. In addition, we identified that the complexity of the co-occurrence network between soil bacteria and fungi varied with geographic location, and both positive and (fewer) negative links were identified. The correlation heatmaps showed different significant positive and negative correlations between the top-ranking taxa of soil bacteria and fungi and soil chemical properties at different latitudes and longitudes. Structural equation modeling (SEM) showed that geographic distribution directly affects and indirectly influences soil chemical properties and microbial communities. Soil chemical properties obviously varied with geographic location, which in turn regulated microbial diversity and influenced the community assembly process of bacterial and fungal communities. Although this approach created a certain understanding of the geographic distribution patterns of inter-root soil microbial communities of *A. senticosus* in northeastern China, the underlying mechanisms of the geographic patterns are not entirely clear. In the future, our research will focus on elucidating the specific mechanisms of ecological processes of community assembly, including deterministic and stochastic mechanisms that shape the geographic patterns of inter-root soil microbial communities in cultivated cyclamen in northeastern China. In addition, we also utilized PICRUSt 2.0 and FUNGuild to predict the potential functions of soil bacterial and fungal microbiomes under different land use patterns, respectively. These findings contribute to a deeper understanding of the geographic patterns of soil microbial communities associated with cultivated *A. senticosus* plants.

## Figures and Tables

**Figure 1 microorganisms-12-02506-f001:**
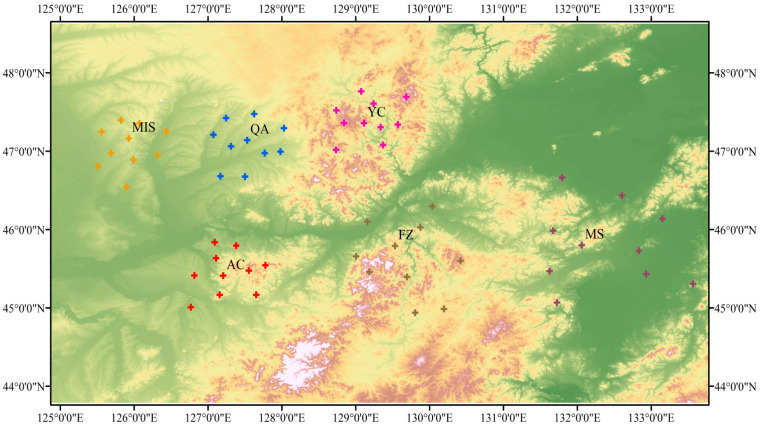
Sampling locations of this study. For each site, ten samples were taken from a region approximately measuring 8250 km^2^, as indicated, and then mixed. MIS: Xiangrong Village, Mingshui County, Suihua City, Heilongjiang Province; FZ: Shuguang Forestry, Fangzheng County, Harbin City, Heilongjiang Province; QA: Qing’an County, Suihua City, Heilongjiang Province; AC: Harbin Yuquan Hunting Ground, Acheng District, Harbin City, Heilongjiang Province; MS: Mishan City, Jixi City, Heilongjiang Province, China; YC: Nanqiao County, Yichun City, Heilongjiang Province.

**Figure 2 microorganisms-12-02506-f002:**
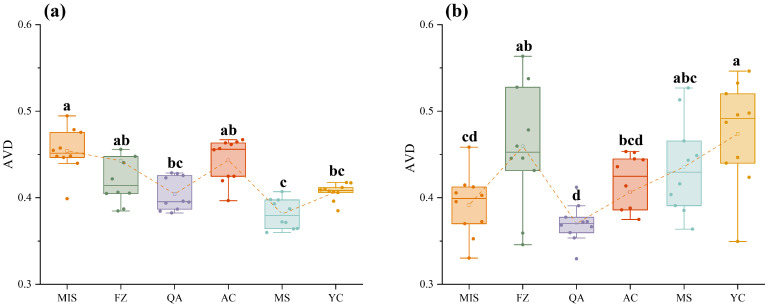
Soil average taxonomic distinctness (AVD) of the bacterial (**a**) and fungal (**b**) communities in the soil sample. Different lowercase letters indicate significant differences in between the sites at the 5% level (*p* < 0.05) as analyzed by ANOVA and Tukey’s post hoc tests.

**Figure 3 microorganisms-12-02506-f003:**
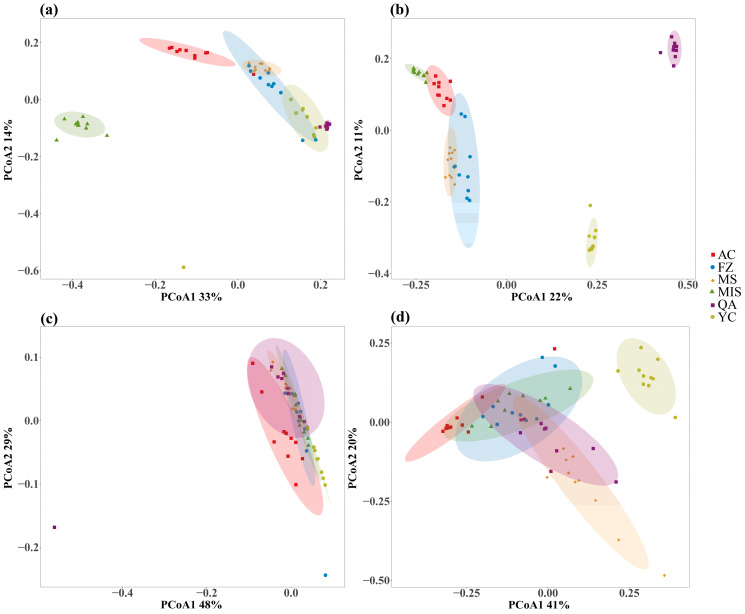
Principal coordinate analysis (PCoA) showing the variances in soil bacterial (**a**) and fungal (**b**) community base on Bray–Curtis distances. Principal coordinate analysis (PCoA) showing the variances in Prediction and Analysis of Soil bacterial (**c**) and fungal (**d**) Community Function base on Bray–Curtis distances.

**Figure 4 microorganisms-12-02506-f004:**
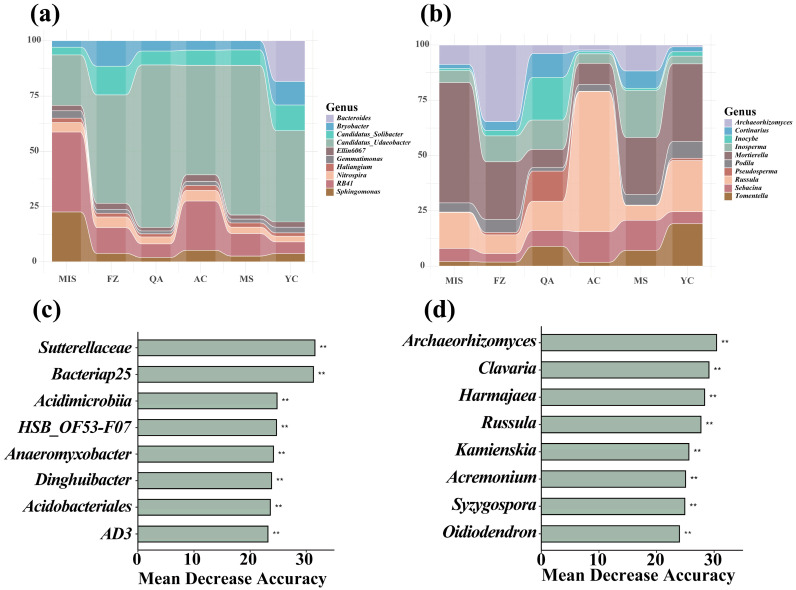
Stacked plots showing the relative abundance of the soil bacterial (**a**) and fungi (**b**) taxa in the soil of the different sites. Random Forest classification modeling identified the bacterial (**c**) and fungal (**d**) taxa that play key roles in these soils. The bars represent variables selected using the classification algorithm. Significance is indicated as ** for *p* < 0.01.

**Figure 5 microorganisms-12-02506-f005:**
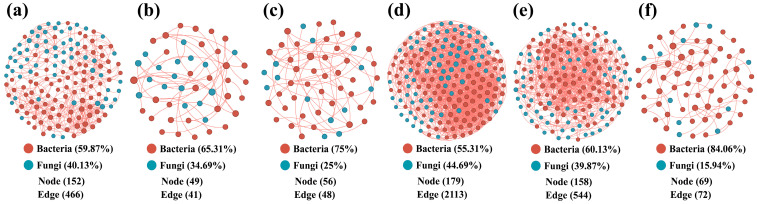
Co-occurrence network showing the complexity of soil bacterial and fungal interactions in the different soils. Different dots in the network represent different OTUs; red dots represent bacteria and blue dots represent fungi. Lines represent interactions with significant correlations. (**a**) MIS; (**b**) FZ; (**c**) QA; (**d**) AC; (**e**) MS; (**f**) YC.

**Figure 6 microorganisms-12-02506-f006:**
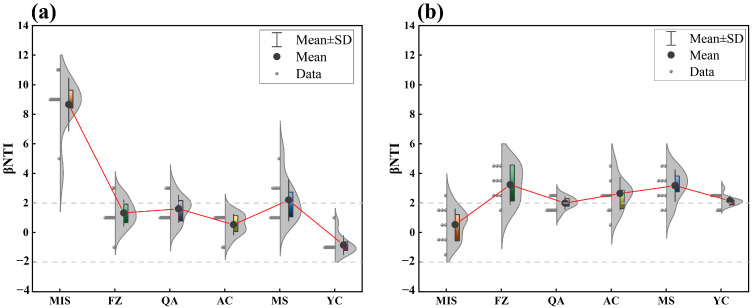
Distribution graph of βNTI (Between-community Nearest Taxon Index) representing ecological processes during community assembly. The βNTI values were determined by contrasting the actual community structure with the predicted theoretical structure based on a null model. If the βNTI value falls within the range of −2 to 2, stochastic processes prevail. However, when the absolute value of βNTI is less than −2 or greater than 2, deterministic processes are prevalent. (**a**) bacteria; (**b**) fungi. The bars of different colors represent different locations. The yellow pillars represent the MIS location, the green pillars represent the FZ location, the purple pillars represent the QA location, the fluorescent yellow pillars represent the AC location, the blue pillars represent the MS location, and the pink pillars represent the YC location.

**Figure 7 microorganisms-12-02506-f007:**
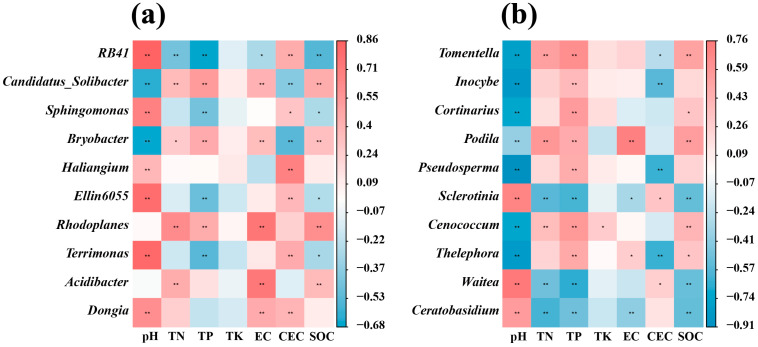
Correlation heatmap showing the correlations between the top ranked bacterial (**a**) and fungal (**b**) genera and soil chemical properties. Red represents a positive correlation and blue a negative correlation. Significance is indicated as * for *p* < 0.05 and ** for *p* < 0.001 based on the Pearson correlation coefficient.

**Figure 8 microorganisms-12-02506-f008:**
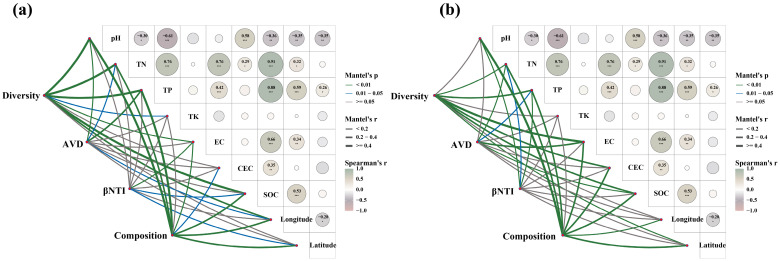
Mantel analysis demonstrating the relationship between soil bacterial (**a**) and fungal (**b**) communities’ diversity, AVD index, βNTI index and composition with soil physicochemical properties, longitude, and latitude. The green and blue lines represent correlations, respectively, and a gray line represents no correlation. The thickness of the line represents the magnitude of the correlation (as per Spearman’s correlation coefficient). Significance is indicated as * for *p* < 0.05, ** for *p* < 0.01 and *** for *p* < 0.001.

**Figure 9 microorganisms-12-02506-f009:**
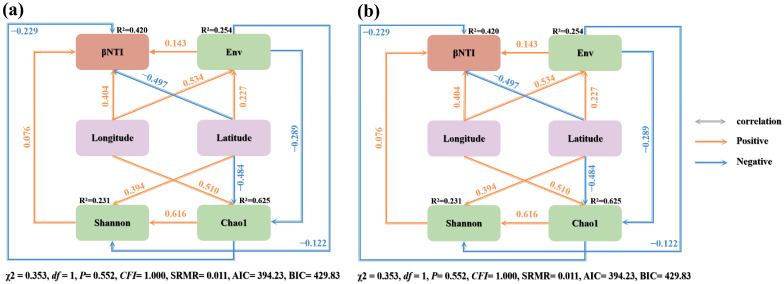
Structural equation modeling (SEM) was used to assess the direct and indirect effects on the bacterial (**a**) and fungal (**b**) communities at the different sample sites. Rectangles represent measured variables; R^2^ represents the proportion of total variance explained for the specific dependent variable. Values above the connecting lines represent the path coefficients, with orange for positive and blue for negative path coefficients (gray: no correlation). Env: Soil chemical properties.

**Figure 10 microorganisms-12-02506-f010:**
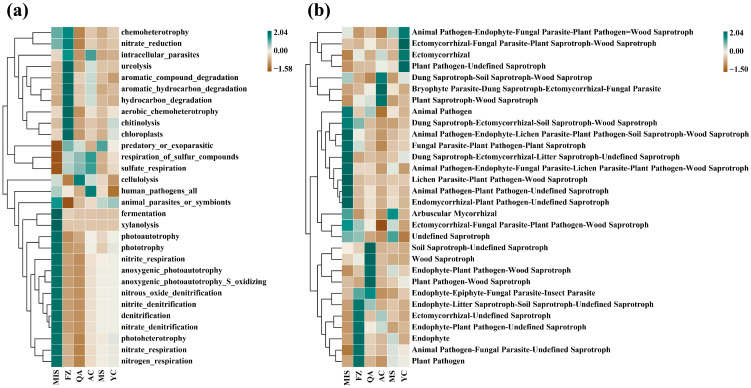
Heatmap of functional enrichment of bacteria for different samples based on PICRUSt2 results (**a**). Heatmap of functional enrichment of fungi in different samples based on FUNGuild results (**b**). The different color blocks indicate the enrichment of different key functions, and the increase in relative abundance is indicated from yellow to green.

**Table 1 microorganisms-12-02506-t001:** Chemical properties of the soil collected at different sites.

Site	SOC (g kg^−1^)	pH	TN (mg kg^−1^)	TP (mg kg^−1^)	TK (g kg^−1^)	EC (mS m^−1^)	CEC (cmol^+^ kg^−1^)
AC	55.1 ± 0.57 ^c^	6.44 ± 0.05 ^b^	0.0059 ± 0.00006 ^b^	795.0 ± 66.45 ^d^	17.0 ± 0.211 ^ab^	159.3 ± 5.63 ^b^	31.7 ± 0.52 ^b^
FZ	29.1 ± 0.57 ^e^	5.53 ± 0.09 ^d^	0.0028 ± 0.00010 ^e^	723.5 ± 50.70 ^e^	16.8 ± 0.395 ^b^	95.9 ± 5.11 ^c^	18.8 ± 0.29 ^f^
MS	60.3 ± 0.73 ^b^	5.79 ± 0.10 ^c^	0.0052 ± 0.00015 ^c^	1060.4 ± 67.04 ^b^	17.1 ± 0.357 ^ab^	89.1 ± 3.00 ^d^	33.1 ± 0.55 ^a^
MIS	19.3 ± 0.50 ^f^	6.94 ± 0.11 ^a^	0.0020 ± 0.00003 ^f^	508.4 ± 26.24 ^f^	16.9 ± 0.543 ^ab^	70.4 ± 4.82 ^e^	30.7 ± 0.42 ^c^
QA	45.9 ± 0.56 ^d^	5.07 ± 0.08 ^f^	0.0039 ± 0.00015 ^d^	932.2 ± 26.21 ^c^	17.3 ± 0.293 ^a^	69.1 ± 6.04 ^e^	24.8 ± 0.26 ^e^
YC	76.5 ± 0.75 ^a^	5.31 ± 0.05 ^e^	0.0061 ± 0.00008 ^a^	1156.0 ± 53.58 ^a^	16.9 ± 0.391 ^ab^	175.5 ± 5.14 ^a^	26.46 ± 0.52 ^d^

TN: total nitrogen; TP: total phosphorus; TK: total potassium; EC: electrical conductivity; CEC: Cation exchange capacity; SOC: Soil organic carbon. Different lowercase letters indicate significant differences (*p* < 0.05).

**Table 2 microorganisms-12-02506-t002:** Alpha diversity indices of the soil bacterial and fungal communities.

Microbial Community	Sample	Shannon Index	Simpson Index	Richness Index	Chao1 Index
Bacteria	AC	7.146 ± 0.2045 ^a^	0.9953 ± 0.0025 ^a^	6568 ± 385.9283 ^ab^	8274.0958 ± 387.4743 ^a^
	FZ	6.989 ± 0.1787 ^a^	0.9961 ± 0.0016 ^a^	6633 ± 633.0136 ^a^	8455.0776 ± 591.7943 ^a^
	MS	6.6361 ± 0.2716 ^a^	0.9826 ± 0.0092 ^a^	6102 ± 393.0759 ^ab^	8089.7406 ± 436.9896 ^a^
	MIS	6.8555 ± 0.203 ^a^	0.9923 ± 0.0058 ^a^	5929 ± 411.8709 ^ab^	7511.4929 ± 440.4228 ^a^
	QA	6.478 ± 0.0821 ^a^	0.9892 ± 0.0016 ^a^	6008.6 ± 293.6454 ^ab^	7871.2933 ± 245.61 ^a^
	YC	6.5037 ± 1.2603 ^a^	0.9753 ± 0.0608 ^a^	5505.8 ± 1798.4429 ^b^	7169.7982 ± 2385.0721 ^a^
Fungi	AC	4.3871 ± 1.2121 ^a^	0.8774 ± 0.2139 ^a^	2156.3 ± 193.8889 ^bc^	2195.0035 ± 233.3214 ^e^
	FZ	4.889 ± 0.7767 ^a^	0.9463 ± 0.055 ^a^	2320.8 ± 194.5975 ^ab^	2800.6151 ± 205.7821 ^c^
	MS	4.8615 ± 0.6511 ^a^	0.9388 ± 0.0779 ^a^	2433.3 ± 95.7695 ^a^	2931.2043 ± 201.4482 ^c^
	MIS	4.8285 ± 0.4692 ^a^	0.9564 ± 0.0338 ^a^	1901.5 ± 125.2945 ^de^	2575.2015 ± 155.2967 ^d^
	QA	4.3839 ± 0.1917 ^a^	0.9571 ± 0.0125 ^a^	1842.7 ± 95.2378 ^e^	2541.2489 ± 163.4956 ^d^
	YC	4.5497 ± 0.4523 ^a^	0.9392 ± 0.033 ^a^	2025.9 ± 259.6486 ^cd^	2329.837 ± 126.5806 ^e^

Different lowercase letters indicate significant differences (*p* < 0.05).

## Data Availability

Certain proprietary or sensitive data that were generated during the course of this research are not publicly available due to confidentiality agreements or ethical considerations. However, these data are available from the corresponding author upon reasonable request and with permission from the relevant authorities. Interested researchers are invited to contact Xin Sui (email: xinsui_cool@126.com) to inquire about accessing these non-public datasets. All data requests will be carefully considered and reviewed to ensure compliance with all applicable laws, ethical guidelines, and data use agreements. This Data Availability Statement serves as a transparent link between the findings presented in this article and the underlying evidence, adhering to Springer Nature’s commitment to research transparency and data accessibility.

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
