# Peer review of "Geographic Distribution Pattern Determines Soil Microbial Community Assembly Process in Acanthopanax senticosus Rhizosphere Soil"

_microorganisms, 2024, doi:10.3390/microorganisms12122506_

Round 1
Reviewer 1 Report
Comments and Suggestions for Authors
Dear Authors,
Review comments are in the PDF file.

Author Response
Response to Reviewer 3 Comments
Dear Editor Celeste Pang,
Dear Reviewer 3,
Thank you very much for giving us the opportunity to revise our manuscript entitled “Geographic distribution pattern determines soil microbial community assembly process in Acanthopanax senticosus rhizosphere soil (microorganisms-3314614)”. Thank you very much for your valuable comments on our MS. According to your comments and suggestions, we carefully revised throughout our MS. Below you will find our point-by-point response.
Comments:
Comments and Suggestions for Authors:
Q1: Present the research objectives clearly to the reader.
Response:
Dear Reviewer,
Thank you for your valuable feedback on our manuscript. We appreciated your suggestion to present the research objectives more clearly to the reader. In response to your comment, we revised the introduction to explicitly state the objectives of our study. The revised sentence read:
"This study aimed to investigate the geographic distribution patterns of soil microbial communities associated with cultivated Acanthopanax senticosus plants in Northeast China, to examine how these patterns varied with geographic location, and to explore the potential drivers and ecological processes underlying these distributions, including soil chemical properties and microbial interactions." (line 16-20)
We believed that this revision provided a clear and concise overview of our research objectives, guiding the reader through the main focus and goals of our study. We hoped that this change addressed your concern and improved the clarity of our manuscript.
Thank you again for your time and consideration. We looked forward to your further feedback.
Comments:
Comments and Suggestions for Authors:
Q2: Avoid repeated words in the manuscript title. Keywords are essential elements for scientific research indexers.
Response:
Dear Reviewer,
Thank you for your insightful comment on our manuscript title. We fully understood the importance of avoiding repeated words, especially in the title, as it not only enhanced readability but also helped in indexing and retrieval by scientific research databases.
In response to your suggestion, we carefully revised the title to eliminate repeated words while retaining the essential keywords for indexing purposes. The new key words:
"Acanthopanax senticosus; rhizosphere microbiome; geographic factors; environmental variables" (line 38-39)
We believed this revised title was more concise, avoided repetition, and still captured the core elements of our research, making it easier for readers and indexers to understand the focus of our study. Thank you again for your valuable guidance. We looked forward to your further feedback on the revised manuscript.
Comments:
Comments and Suggestions for Authors:
Q3: Which substances from Acanthopanax senticosus are used in medicine?
Is this plant used in traditional and ancient medical practice or in contemporary medical practice (medicines)?
Put this in context.
Response:
Dear reviewer,
Thank you for your inquiry about the medicinal uses of Acanthopanax senticosus. This plant, commonly known as Siberian ginseng or Acanthopanax senticosus, was indeed valued for its various bioactive compounds that were utilized in both traditional and contemporary medical practices. In traditional medicine, particularly in Chinese and Russian folk medicine, Acanthopanax senticosus had been used for centuries to enhance physical stamina, improve mental clarity, and support the immune system. The root and bark of the plant were typically employed, as they were rich in compounds such as Acanthopanax senticosus, which were believed to contribute to its medicinal properties. In contemporary medical practice, Acanthopanax senticosus gained attention for its potential health benefits, including its adaptogenic properties, which helped the body cope with stress. Extracts from the plant were studied for their immune-boosting, anti-fatigue, and anti-inflammatory effects. These extracts were often incorporated into dietary supplements, herbal remedies, and even some pharmaceutical products aimed at improving overall well-being and resilience.To summarize, Acanthopanax senticosus contained various bioactive substances, particularly eleutherosides, that were used in both traditional and modern medical contexts. Its applications ranged from enhancing physical and mental performance in traditional practices to being a component in contemporary supplements and medications designed to support health and stress resistance. The specific changes are as follows: ‘
“In traditional Chinese Medicine, Acanthopanax senticosus has high medicinal value with anti-fatigue, anti-inflammatory and cardiovascular activity [21]. It has been widely cultivated in the past decades and its growth is stimulated by inter-root soil microorganisms [22]. A. senticosus is rich in a variety of compounds with notable pharmacological effects, among which eleutherosides exhibit a bidirectional regulatory function on the central nervous system [12], effectively enhancing the body's anti-fatigue capability; polysaccharides components are capable of boosting immunity; and volatile oils possess marked antibacterial, anti-inflammatory [13], and sedative effects. These constituents collectively contribute to the extensive medicinal value of A. senticosus. ” (line 90-104)
I hope this information provided a comprehensive context for the medicinal uses of Acanthopanax senticosus. If you had any further questions, you were free to ask.
Comments:
Comments and Suggestions for Authors:
Q4: What is the technical name used to designate the microorganisms that inhabit the root-soil interface?
Response:
Dear reviewer,
In response to your inquiry regarding the technical name used to designate the microorganisms that inhabited the root-soil interface, I would like to clarify that there was no single, specific technical term that exclusively encompassed all such microorganisms. Rather, this field of study relied on a variety of research methods and techniques to investigate the diverse microbial communities found in the rhizosphere. From a broad and encompassing perspective, the term "rhizosphere microbiome" was often used to refer to the collection of microorganisms that inhabited this unique environment. Alternatively, when focusing on the processes of isolating and identifying these microorganisms, one might have used the term "rhizosphere microbial isolation and identification techniques." It is important to note that the study of rhizosphere microorganisms involved a multidisciplinary approach, incorporating techniques from microbiology, molecular biology, ecology, and other related fields. Researchers employed a range of methods, including sampling techniques, cultivation-based approaches, molecular fingerprinting, and high-throughput sequencing, to study the composition, function, and dynamics of rhizosphere microbial communities.
Comments:
Comments and Suggestions for Authors:
Q5: Are there changes in the structures and tissues of plant roots during the establishment of symbiosis?
Response:
Dear reviewer,
Thank you very much for your input. In the process of establishing symbiotic relationships, changes occurred in the structures and tissues of plant roots. Therefore, we have revised the content of the article accordingly. The specific revisions are as follows:
“Soil microorganisms form a close symbiotic relationship with the root system of these plants, building a complex ecosystem that exerts a profound influence on the plant’s growth [23]. During this process, the structures and tissues of the plant roots undergo notable changes to facilitate this interaction.” (line 101-104)
Thank you again for your valuable guidance. We looked forward to your further feedback on the revised manuscript.
Comments:
Comments and Suggestions for Authors:
Q6: Rhizosphere microbial community
Response:
Dear reviewer,
Thank you very much for your feedback. We have made revisions to that part of the article. The specific revised content is as follows:
“However, few studies have explored the geographic distribution patterns and community assembly process of Rhizosphere microbial community.” (line 108-111)
Comments:
Comments and Suggestions for Authors:
Q7: Present the research objectives clearly to the reader.
Response:
Dear reviewer,
Thank you for your valuable feedback on the importance of clearly presenting research objectives. I appreciate your guidance in ensuring that our research is communicated effectively.
In response to your suggestion, I have revised the introduction of our research paper to explicitly outline our objectives. Here is a clear and concise presentation of our research goals:
“To investigate the differences in soil microorganisms associated with the root systems of A. senticosus cultivated at various sites in Northeast China, we conducted a comprehensive analysis of the inter-root soil microorganisms, with the aim of exploring the potential impact of these variations on the growth and health of A. senticosus.” (line 112-117)
Comments:
Comments and Suggestions for Authors:
Q8: Provide the soil classification for each collection site. Use the World Reference Base for Soil Resources (WRB), Soil Taxonomy Staff USDA or the Chinese Soil Classification System.
Add information about the prevailing climate at each collection site (rainfall, minimum, average and maximum temperatures, etc.).
Response:
Dear reviewer,
Thank you for your suggestions on the classification of soil properties. Based on your suggestions, we have made modifications as follows:
“Yichun City's Nancha County is characterized by a temperate continental climate, with annual precipitation ranging from 550 to 750 millimeters, mostly occurring in July and August. The soil type in this area is classified as leached soil. Fangzheng County, on the other hand, experiences a cold temperate continental monsoon climate and has an annual average precipitation of 579.7 millimeters. The soil here is also classified as leached soil. Acheng District features a cold temperate continental monsoon climate, with June being the hottest month, averaging 24.2°C, and the extreme maximum temperature reaching 38.4°C. Similarly, the soil type in this region is leached soil. Jixi City, to which Mishan City belongs, is located on the east coast of the Asian continent at mid-latitudes and has a temperate continental monsoon climate, with average temperatures ranging from 3.5°C to 4.2°C. The soil type here is albic soil. Qing'an County experiences a cold temperate continental monsoon climate, with an annual average temperature of 1.69°C, a frost-free period of approximately 128 days, and an annual average precipitation of 577 millimeters. The soil in this area is classified as black soil. Mingshui County has a temperate continental climate, with January being the coldest month, averaging -20.9°C, and July being the hottest, averaging 21.9°C. The annual temperature range is 42.8°C, and the annual average temperature is 2.0°C. The soil type in this region is red soil”. (line 135-152)
Comments:
Comments and Suggestions for Authors:
Q9: This topic is too long.
I suggest that authors present a brief description and provide a detailed description as supplementary material.
Response:
Dear reviewer,
In response to your comment, we would like to clarify that we expanded this section based on feedback from previous reviewers who had requested more detailed information. We believed that the additional content provided crucial context and depth to our analysis, which was why we retained it in the main text. However, recognizing the importance of balancing detail with readability, we considered your suggestion during our subsequent submission and explored the possibility of moving some of the more extensive details to a supplementary table or appendix. This allowed readers to access in-depth information as needed while maintaining clarity and concision in the main body of the text. Thank you again for your constructive feedback; we looked forward to incorporating your suggestions to enhance the overall quality of our work.
Comments:
Comments and Suggestions for Authors:
Q10: At least this information must be in the main text as a table.
Response:
Dear reviewer,
Thank you for your suggestion regarding the placement of the relevant information. We have carefully considered your feedback and made the necessary adjustments. In accordance with your comments, we have now moved the pertinent information to Table 1 in the main text. We believe that this change will enhance the readability and accessibility of the data, making it easier for readers to grasp the key points. Once again, we appreciate your valuable input and thank you for your attention to detail. We look forward to any further suggestions you may have for improving our work.
Comments:
Comments and Suggestions for Authors:
Q11: Alpha diversity indices should be presented as tables in the main text.
Response:
Dear reviewer,
Thank you for your suggestion on the presentation of alpha diversity indices. We took your feedback to heart and made the necessary revisions. In response to your comment, we created a new table presenting the alpha diversity indices and placed it as Table 2 in the main text. This provided a clearer and more accessible overview of the data for our readers. We appreciated your constructive input and looked forward to any further suggestions you might have to improve our work.
Comments:
Comments and Suggestions for Authors:
Q12: Does this type of analysis assess alpha or beta diversity?
Response:
Dear reviewer,
Thank you for your inquiry regarding the type of diversity assessed by Principal Coordinate Analysis (PCoA) in our study. To clarify, PCoA was indeed used to assess beta diversity. This analysis evaluated the similarity in composition of the soil bacterial and fungal communities across different samples, focusing on the differences between them rather than the diversity within a single sample, which would be alpha diversity. We hoped this answered your question. If you had any further inquiries or suggestions, we would have been happy to address them.
Comments:
Comments and Suggestions for Authors:
Q13: Improve the resolution of the figure. The letters indicating the panels (a, b, c and d) are disproportionately large in the figure. Adjust the size of these letters.
Response:
Dear reviewer,
Thank you for your feedback on the figure in our submission. We took note of your comments regarding the resolution and the size of the letters indicating the panels (a, b, c, and d). In response, we made the necessary adjustments. We resized the letters to ensure they were proportionally appropriate and did not overpower the rest of the figure. Additionally, we imported a higher resolution version of the photograph to enhance the overall clarity and detail of the figure. We appreciated your constructive input and believed these changes improved the readability and effectiveness of the figure in conveying our results. If you had any further suggestions or required additional modifications, we would have been happy to address them.
Comments:
Comments and Suggestions for Authors:
Q14: All these results seem to be accessory (supplementary).
Start the topic with the key result (lines 377-388).
The text in the current presentation does not arouse the reader's curiosity. Since it starts with supplementary results. To "capture" the reader's attention, one should present the key results whenever starting a topic.
Response:
Dear reviewer,
We sincerely appreciated your invaluable feedback on the structure of our presentation. We were grateful for your suggestion to prioritize key findings when introducing a new topic, as it indeed enhanced reader engagement and piqued curiosity. In accordance with your advice, we rearranged the text to highlight the key results, specifically by moving the section discussing the key findings (lines 377-388) to the beginning of the topic to ensure it immediately captured the readers' attention. The supplementary results followed, providing additional context and support for the key findings. We believed that this new structure would make the presentation more compelling and easier to comprehend. The revised content was as follows:
“Microbial co-occurrence networks were analyzed at the phylum level for both bacteria and fungi (Fig. S4). Notably, the bacterial networks were complex in four out of six soil communities (Fig. S4a-i), with the soil from YC exhibiting the highest number of bacterial nodes (281) and edges (35). In all networks, the majority of bacterial interactions were positive (81-93%), and in YC, only positive interactions were observed (Fig. S4; Table S4). The phylum Acidobacteriota ranked first in four soil samples, while Proteobacteria ranked highest in MS and Verrucomicrobiota in AC. Acidobacteriota and Proteobacteria combined represented 51-54% of the nodes in the more complex bacterial networks.
In fungal co-occurrence networks, Ascomyeota consistently ranked first. The fungal networks (Fig. S4g-l) were simpler than the bacterial networks, with 6 to 43 nodes and 3 to 22 edges. The fungal network of UC was the simplest, while FZ's was the most complex. Only FZ and MIS soil fungal networks contained both positive and negative interactions; others had only positive interactions. When considering bacterial-fungal mutualistic relationships (Fig. 5), the network was most complex in AC soil (179 nodes, 2113 edges), followed by similarly complex networks in MS and MIS. The networks in FZ, QA, and YC were the simplest, indicating weaker bacterial-fungal mutualistic relationships in these soils.” (line 421-438)
Thank you once again for your insightful suggestions. Should you have any further comments or recommendations, we would be delighted to hear them.
Comments:
Comments and Suggestions for Authors:
Q15: Which rank value (-2 to 2) represents the stochastic and deterministic processes?
It might be interesting to add the scale.
Response:
Dear reviewer,
Thank you for your inquiry regarding the interpretation of βNTI values in the context of stochastic and deterministic processes. Indeed, it was a fascinating aspect of microbial ecology that provided deep insights into the driving forces behind community assembly. As you mentioned, βNTI values that ranged from -2 to 2 were indicative of stochastic processes. This meant that within this range, the assembly of microbial communities was largely influenced by random events, such as dispersal limitation, ecological drift, or other unpredictable factors. Stochastic processes played a significant role in shaping microbial community structure when environmental conditions were relatively homogeneous or when species interactions were weak. On the other hand, βNTI values that fell outside this range, either less than -2 or greater than 2, suggested the presence of deterministic processes. When βNTI was less than -2, it indicated that species in the community were more phylogenetically dispersed than expected by chance, which might have been due to environmental filtering or other factors that selected for distantly related species. Conversely, when βNTI was greater than 2, it suggested that species were more phylogenetically clustered than expected, indicating strong competitive interactions or niche partitioning among closely related species. Understanding the balance between stochastic and deterministic processes was crucial for predicting how microbial communities responded to environmental changes, such as climate change or pollution. By quantifying βNTI values, researchers gained a better understanding of the underlying mechanisms driving community assembly and used this information to inform conservation strategies or management practices. I hope this explanation clarified the relationship between βNTI values and the stochastic and deterministic processes in microbial community assembly. If you had any further questions or needed additional information, please didn't hesitate to ask.
Comments:
Comments and Suggestions for Authors:
Q16: Insert the climatological information of the collection sites. Present the discussion based on the climatic variations.
Response:
Dear reviewer,
Thank you for your insightful suggestion regarding the inclusion of climatological information for our collection sites and the subsequent discussion based on climatic variations. I have taken your advice to heart and have made the necessary revisions to enrich our analysis.
Below is an expanded section of the paper that now incorporates detailed climatological information and a discussion of its implications:
“This location creates specific climatic and ecological conditions that may not be conducive to the survival of a wide range of fungi, which in turn affects microbial community structure and diversity in the soil [67,68]. The lower temperatures at the QA site provide a moderate environment that falls between the extremes of arid and tropical climates, fostering the growth of certain fungal species adapted to these moderate temperatures while potentially limiting the proliferation of fungi that thrive in hotter or colder environments.” (line 589-595)
Comments:
Comments and Suggestions for Authors:
Q17: What environmental conditions are most favorable to the fungal community?
Are climate variations alone sufficient to explain the greater diversity of fungi at site QA?
Response:
Dear reviewer,
Thank you for your valuable suggestions. We carefully revised our content based on your feedback to ensure clarity and consistency.
In responding to your question about the environmental conditions most favorable to the fungal community, we found that a combination of factors played a crucial role. These included moderate temperatures, consistent moisture levels, and a rich supply of organic matter in the soil. Furthermore, the pH level and texture of the soil, as well as the presence of certain host plants, also influenced the growth and diversity of fungi. Therefore, we made the necessary changes to the article content accordingly. The following is the modified content:
”When environmental conditions are favorable for a wide range of fungi, the ecology is relatively stable and lacks the dramatic environmental changes that drive fungal community diversification [69,70]. However, it is important to note that climate variations alone may not be sufficient to fully explain the greater diversity observed at site QA; other environmental factors, such as soil type, vegetation cover, and disturbance history, may also play crucial roles in shaping the fungal community structure.” (line 597-603)
Comments:
Comments and Suggestions for Authors:
Q18: Check the grammar.
Response:
Dear reviewer,
Thank you for your inquiry regarding the grammar of the sentence. We have reviewed and revised it accordingly. The specific changes are as follows:
“The AVD Index is a key indicator that describes the stability of the soil microbial community within the root zone of A. senticosus, and this stability is essential for assessing ecosystem health and predicting the impacts of environmental change on microbial communities [71].” (line 606-609)
We hope this revision meets your requirements. If you have any further questions or need additional assistance, please do not hesitate to contact us.
Comments:
Comments and Suggestions for Authors:
Q19: Do all collection sites have the same soil class?
Does the soil have the same textural gradient across the different collection sites?
Does the soil have the same chemical properties across the different collection sites?
Which soil properties have the greatest influence on the assembly of the soil microbial community?
Response:
Dear reviewer,
In response to the inquiries regarding soil class, textural gradient, chemical properties, and their influence on soil microbial community assembly across different collection sites, the following revised paragraph is provided:
“The soil class, textural gradient, and chemical properties may vary across different collection sites. The observed differences in soil properties could be related to the high latitude and longitude of sites QA and YC, as well as the specific climatic and environmental conditions, such as temperature and humidity, that are prevalent in these locations. These conditions may have indirectly contributed to the accumulation of soil organic carbon (SOC). Higher SOC levels indicate fertile soil rich in organic matter, which serves as an adequate food source for microorganisms [74]. Among these soil properties, SOC content, along with other factors such as pH, moisture, and nutrient availability, likely has a significant influence on the assembly and diversity of the soil microbial community.” (line 616-625)
Comments:
Comments and Suggestions for Authors:
Q20: In what temperature, humidity and pH ranges does this genus have wide abundance?
Response:
Dear reviewer,
Regarding the genus Candidatus_Udaeobacter in soil environments, there was no definitive information on the specific temperature, humidity, and pH ranges within which it had wide abundance. However, based on available research, Candidatus_Udaeobacter was identified as a dominant bacterial genus in the rhizosphere soil of various plants. For example, it was found to be prevalent in the soil surrounding tea trees and medicinal plants such as Lamiophlomis rotata on the Tibetan Plateau. These soil environments likely spanned a range of temperatures, humidities, and pH levels depending on the specific location, climate, and soil type. While specific environmental thresholds for optimal abundance of Candidatus_Udaeobacter in soil were not well-documented, its prevalence in diverse soil environments suggested that this genus might be adaptable to a range of conditions. Further research, particularly controlled studies examining the growth and abundance of Candidatus_Udaeobacter in soil under different temperature, humidity, and pH conditions, would have been needed to determine precise thresholds for its optimal growth and activity. In summary, while definitive ranges for Candidatus_Udaeobacter abundance in soil were not available, its identification as a dominant genus in various soil environments indicated its adaptability to different environmental conditions.
Comments:
Comments and Suggestions for Authors:
Q21: The explanation of community complexity as a function of longitude and latitude is not adequate.
There are numerous other factors in the landscape that have a greater impact on the assembly of the rhizosphere and soil microbiome.
Response:
Dear reviewer,
Thank you for your insightful comments and constructive suggestions. We fully acknowledge that our initial explanation of community complexity as a function of longitude and latitude was inadequate, and we appreciate your guidance in this matter. Upon reflecting on your feedback, we realized that numerous other factors in the landscape indeed have a significant impact on the assembly of the rhizosphere and soil microbiome. Therefore, we have revised our article to incorporate a more comprehensive analysis of these factors. We now discuss the influence of soil type, climate, vegetation, land use history, and other environmental variables on the structure and diversity of the rhizosphere and soil microbiome. We have made these changes to ensure that our article provides a more accurate and thorough understanding of the complex interplay between geographical location and other environmental factors in shaping microbial communities. We hope that these revisions address your concerns and enhance the overall quality of our work. The following revised paragraph is provided:
“The bacterial and fungal co-occurrence network interactions surrounding the roots of A. senticosus were most complex at the AC location (Fig. 5). While this site is located at a lower latitude and (except for MIS) at a lower longitude, it is important to acknowledge that longitude and latitude alone do not fully explain the complexity of the rhizosphere and soil microbiome. Numerous other factors in the landscape, such as soil type, climate, vegetation, and anthropogenic activities, also have significant impacts on the assembly and diversity of these microbial communities. The high diversity of bacterial and fungal species at the AC location, which results in a richness of potential interactions and increases the complexity of the identified networks, is likely influenced by a combination of these various factors.” (line 741-750)
Comments:
Comments and Suggestions for Authors:
Q22: The supplementary material file is unavailable for reading.
So far, I have not been able to check any information in the supplementary material. This is because the file is corrupt and/or ineligible.
Response:
Dear reviewer,
I sincerely apologize for any inconvenience caused by the unavailability of the supplementary material file. Upon identifying the issue, we promptly took steps to resolve it and have now re-uploaded a new and correct version of the supplementary material. We deeply regret any inconvenience this may have caused and greatly appreciate your patience and understanding. Should you encounter any further issues or have any questions, please do not hesitate to contact us.
Comments:
Comments and Suggestions for Authors:
Q23: The discussion subjectively addresses the effects of spatial variation and some soil properties.
Suggestion: discuss the results objectively.
The manuscript is long. It is not attractive to the reader to read.
There are repeated discussions throughout the topic. (Always the same factors modulate different variables).
Response:
Dear reviewer,
Thank you for taking the time to review our manuscript and providing valuable feedback. We appreciated your constructive comments and suggestions for improvement. Regarding your concern about the subjective discussion of the effects of spatial variation and soil properties, we acknowledged the importance of maintaining an objective tone in scientific writing. We revised the discussion section to ensure it presented the results in a more objective and data-driven manner, focusing on the empirical findings and their implications. We understood that the length of the manuscript may have made it less attractive to readers. In response, we carefully reviewed and edited the text to eliminate unnecessary repetitions and streamline the narrative. Our goal was to present a concise and engaging manuscript that effectively communicated our research findings. Furthermore, we recognized the issue of repeated discussions throughout the topic. To address this, we reorganized the manuscript to avoid redundancy and ensure that each section contributed unique and valuable information to the overall narrative. We also made sure to highlight the distinct contributions of different factors to the variables discussed, rather than repeating the same points. The revised content is as follows:
“It is noteworthy that the co-occurrence networks of soil bacteria and fungi in the YC region exclusively comprise positive relationships (Fig. S4), a unique phenomenon closely linked to niche differentiation. The specific latitude and longitude of the region, along with its distinctive climatic conditions [110], soil characteristics, and biogeographic patterns [111,112], not only provide abundant nutrients and ideal growth environments for microorganisms but also facilitate niche differentiation among them. The low pH and high levels of total nitrogen (TN), total phosphorus (TP), electrical conductivity (EC), and soil organic carbon (SOC) in YC soils enable different types of microorganisms to select and occupy the most suitable niches based on their environmental adaptability, thereby reducing interspecies competition and promoting community stability and harmony [7]. Furthermore, the unique biogeographic pattern of high latitude and longitude regions offers a relatively stable ecological environment conducive to the coexistence of specific bacteria and fungi, and niche differentiation further minimizes direct competition among them, thus maintaining this positive relationship. Niche differentiation not only enhances microorganisms' resource utilization efficiency but also strengthens community stability and diversity, serving as one of the crucial factors contributing to the positive correlations observed in the soil microbial co-occurrence networks of the YC region.” (line 806-823)
Thank you again for your feedback. We were committed to improving the quality and clarity of our manuscript and looked forward to submitting a revised version that addressed your concerns.
Comments:
Comments and Suggestions for Authors:
Q24: The cultivation history of A. senticosus is unknown. The authors do not provide sufficient information about the areas where A. senticosus is cultivated.
The response of the microbial community also depends on the cultivation history or land use history. There are a multitude of factors that can modulate the soil and rhizosphere microbiome.
Perhaps the authors could provide information on each collection area (detailed characterization). The authors could make the discussion more succinct and objective.
Response:
Dear reviewer,
Thank you for your thoughtful comments and suggestions regarding our article. We appreciate your pointing out that the cultivation history of A. senticosus was not adequately addressed and that the response of the microbial community is influenced by various factors, including cultivation or land use history. In response to your feedback, we have revised our article to provide more detailed information about the areas where A. senticosus is cultivated. We now include a detailed characterization of each collection area, highlighting the cultivation history, soil type, climate, and other relevant environmental factors. This information is crucial for understanding the potential variations in the soil and rhizosphere microbiome across different sites. Furthermore, we have refined our discussion to make it more succinct and objective. We now emphasize the multitude of factors that can modulate the soil and rhizosphere microbiome, including cultivation history, and discuss how these factors may interact to influence the microbial community structure. We believe that these revisions address your concerns and enhance the overall quality and clarity of our work. Thank you once again for your valuable input. Should you have any further questions or suggestions, please feel free to reach out. The specific changes are as follows:
“We found that for the MIS site, which has a unique cultivation history of Acanthopanax senticosus, the soil fungal community assembly processes differed significantly from those in the other five sites (Fig. 6; Fig. S5). Stochastic processes predominated in the MIS area, potentially influenced by the specific environmental conditions associated with the site's location and its cultivation practices. Given that fungal spores possess strong dispersal abilities and can be widely spread by various means, including wind, water, and animals, the particular geographic conditions at MIS, combined with its cultivation history, may have facilitated the random distribution of spores, resulting in a community structure dominated by stochasticity [118,119]. Furthermore, the soil fungi in the MIS area, which has been specifically cultivated for Acanthopanax senticosus, may not have established strong competitive or symbiotic relationships with each other. This could be due to the relatively recent introduction or cultivation of Acanthopanax senticosus, which might have disrupted pre-existing fungal relationships or introduced new fungal species that are still adapting to the environment. As a result, stochastic factors play a more prominent role in community formation [120].” (line 851-865)
Comments:
Comments and Suggestions for Authors:
Q25: I suggest that the authors merge this topic - both results and discussion - with topic 3.1 and 4.1.
Response:
Dear reviewer,
Thank you for your thoughtful suggestion to merge the topic on soil fungal community assembly processes at the MIS site, particularly in relation to Acanthopanax senticosus, with topics 3.1 and 4.1. We appreciated your efforts to help streamline and enhance the coherence of our manuscript. Upon careful consideration of your suggestion, we reviewed the content of all relevant sections, including 4.1 (Soil Microbial Diversity and Composition) and 4.3 (Key Regulators Influencing Changes in Soil Microbial Communities). While we understood the potential benefits of merging similar topics, we believed that the distinct nature of the soil fungal community assembly processes at the MIS site, especially in the context of Acanthopanax senticosus cultivation, warranted a separate and detailed discussion. The section on soil fungal community assembly processes delved into specific environmental conditions, dispersal mechanisms, and succession stages that were unique to the MIS site. These aspects, while related to the overall microbial diversity and composition as well as key regulators, required a dedicated space to fully explore their implications and significance. Therefore, after thorough deliberation, we decided to maintain the current structure of our manuscript, keeping the discussion on soil fungal community assembly processes as a separate topic. We believed this approach would allow for a more in-depth analysis and clearer communication of our findings. Thank you once again for your valuable input. We appreciated your understanding and continued support in refining our work.
Comments:
Comments and Suggestions for Authors:
Q26: Soil Microbial Community
Response:
Dear reviewer,
Thank you very much for your feedback. Based on your suggestions, we have made modifications to the content. The specific changes are as follows:
“4.4. Prediction and Analysis of Soil Microbial Community Function” (line 1000)
Comments:
Comments and Suggestions for Authors:
Q27: The manuscript is very long. I suggest that the authors present the results and discussion in a direct and objective manner.
Current publishing models establish "shorter and more direct" manuscripts. This manuscript is 30 pages long. There are 24 pages of authorial content. And a huge supplementary material (806,468 KB or 0.8 GB).
Response:
Dear reviewer,
Thank you for your constructive feedback on our manuscript. We fully understood the importance of presenting results and discussions in a direct and objective manner, especially given the current publishing models that favored shorter and more concise manuscripts. In response to your suggestions, we thoroughly revised our manuscript to ensure that the content was more streamlined and focused. We condensed the text, eliminated redundant information, and restructured certain sections to enhance clarity and readability. As a result, we successfully reduced the length of the manuscript while preserving the essential findings and conclusions. Furthermore, we also took note of the size of the supplementary material. To address this concern, we reviewed and optimized the content of the supplementary material to ensure it was more manageable in size. We removed any unnecessary files and compressed the remaining ones to significantly reduce the overall file size. We uploaded a revised version of the manuscript along with a new, more concise supplementary material. We believed that these changes would make the manuscript more accessible and appealing to readers and reviewers. Thank you once again for your valuable feedback. We appreciated your time and effort in helping us improve our work.
We sincerely appreciate your valuable time and expertise in reviewing our manuscript. Your insightful comments and suggestions have been incredibly helpful in improving the quality of our work. The issues you raised are highly professional and objective, and they are of great importance to us. In the process of contemplating, addressing, and responding to the points you highlighted, we have learned a lot and gained considerable insight. This has also been immensely beneficial for our future work endeavors. We hope that the revised manuscript now meets your expectations and the standards of the journal. Thank you once again for your invaluable contribution to our research. We look forward to your further feedback. Taking this opportunity, I, on behalf of my co-authors and myself, would like to express our most sincere blessings to you. May your days be filled with joy, your work with satisfaction, and your life with fulfillment!
Best Regards
Yours Sincerely,
Dr. Xin Sui
Engineering Research Center of Agricultural Microbiology Technology, Ministry of Education & Heilongjiang Provincial Key Laboratory of Ecological Restoration and Resource Utilization for Cold Region & Key Laboratory of Microbiology, College of Heilongjiang Province & School of Life Sciences, Heilongjiang University, Harbin 150080, China. xinsui_cool@126.com (X. Sui).
Reviewer 2 Report
Comments and Suggestions for Authors
The manuscript, titled "Geographic Distribution Pattern Determines Soil Microbial Community Assembly Process in Acanthopanax senticosus Rhizosphere Soil," explores microbial diversity, composition, and community assembly in the rhizosphere of Acanthopanax senticosus in Northeast China. Using high-throughput sequencing and bioinformatics, the study identifies how geographic location, soil chemistry, and stochastic versus deterministic processes shape bacterial and fungal communities. Through advanced modeling and network analyses, this research offers valuable insights into microbial community dynamics related to plant growth and soil health.
1. Adding a brief comparison with studies on other medicinal plants could improve the context of the study’s contributions.
2. Citing studies on microbial biogeography in similar ecosystems would strengthen the rationale for focusing on A. senticosus in Northeast China. References to research on geographic microbial patterns along latitudinal gradients could enhance the background.
3. Clarifying data accessibility by depositing sequence data in an accessible repository, such as NCBI SRA, would support reproducibility.
4. The paper should discuss why certain interactions are prevalent at specific sites, potentially due to environmental factors or niche competition.
5. The co-occurrence network analysis effectively illustrates microbial interactions, but including a comparison with null models or validation steps would help confirm that observed associations are non-random. Incorporating these, as recommended in similar studies, would improve the robustness of the network analysis and provide stronger support for interpreting microbial relationships.
Author Response
Response to Reviewer 1 Comments
Dear Editor,
Dear Reviewer 1,
Thank you very much for giving us the opportunity to revise our manuscript entitled “Geographic distribution pattern determines soil microbial community assembly process in Acanthopanax senticosus rhizosphere soil (microorganisms-3314614)”. Thank you very much for your valuable comments on our MS. According to your comments and suggestions, we carefully revised throughout our MS. Below you will find our point-by-point response.
Comments:
Comments and Suggestions for Authors :
Q1: Adding a brief comparison with studies on other medicinal plants could improve the context of the study’s contributions.
Response 1:
Dear reviewer,
Thank you for your considerable suggestion for our manuscript and we have now included a brief comparison with relevant studies on other medicinal plants. The specific modifications are as follows:
“Studies on similar plants, such as Panax ginseng and Glycyrrhiza uralensis, have demonstrated that rhizosphere microorganisms significantly influence plant nutrient uptake and stress tolerance [11].” (line 84-86)
“However, few studies have explored the geographic distribution patterns and community assembly process of rhizosphere microorganisms in A. senticosus. Understanding these dynamics can provide insights into the ecological adaptations of A. senticosus and its interactions with soil microbes.” (line 91-95)
Q2: Citing studies on microbial biogeography in similar ecosystems would strengthen the rationale for focusing on A. senticosus in Northeast China. References to research on geographic microbial patterns along latitudinal gradients could enhance the background.
Response2:
Dear reviewer,
Thank you for your valuable comments for my manuscript. We have added the incorporated reference to studies on microbial biogeography in similar ecosystem. The specific modifications are as follows:
“Studies on microbial biogeography have demonstrated that geographic factors, such as latitudinal gradients and soil chemistry, play a pivotal role in shaping microbial community composition and diversity [8,9]. For example, deterministic processes like niche differentiation and species sorting often interact with stochastic factors, influ-encing the distribution of microbial taxa across latitude [10,11]. However, drivers remain poorly understood in ecosystems associated with medicinal plants, where rhizosphere microorganisms are integral to plant health and productivity. Investigating microbial community assembly in the rhizosphere of different medicinal plants can uncover how plant-microbe interactions evolve under distinct ecological pressures.” (line 52-60)
Q3: Clarifying data accessibility by depositing sequence data in an accessible repository, such as NCBI SRA, would support reproducibility.
Response3:
Dear reviewer,
We sincerely appreciated your invaluable feedback on the importance of depositing sequence data in an accessible repository such as NCBI SRA to support research reproducibility. We fully recognized the significance of making our data publicly available and ensuring transparency in the research process. We were delighted to inform you that we were in the process of uploading our sequence data to NCBI SRA. Upon completion of the upload and acquisition of the NCBI accession numbers, we promptly contacted the editor to supplement this information in our publication. We understood the crucial role that providing these details played in ensuring the reproducibility of our findings and assured you of our commitment to adhering to best practices in data sharing. Thank you once again for your suggestion, and we eagerly looked forward to sharing the NCBI accession numbers with you and the broader scientific community in the near future.
Q4: The paper should discuss why certain interactions are prevalent at specific sites, potentially due to environmental factors or niche competition.
Response 4:
Dear reviewer,
In response to your feedback highlighting that the prevalence of certain interactions in specific sites might be attributed to environmental factors or niche competition, we revised our paper to include a comprehensive discussion on this topic. We analyzed the environmental factors and niche competition that could potentially drive the ubiquity of certain interactions in particular locales, providing detailed explanations and supporting evidence. We believed that this addition not only strengthened our paper but also enhanced its contribution to the field. The specific modifications are as follows:
“The complexity of bacterial and fungal co-occurrence networks surrounding the roots of A. senticosus at the AC site can be attributed to the unique ecological environment that promotes niche differentiation and competition among diverse microbial species. Influenced by its geographical location (lower latitude and lower longitude), the soil environment likely harbors a more diversified array of microhabitats [94], providing ample living space and resources for different bacterial and fungal species [95]. However, the limited availability of resources fuels intense competition among microorganisms, each striving to occupy the most favorable niche for itself [96]. In this competitive landscape, various microbial species develop unique metabolic pathways and ecological functions to reduce direct competition with others, thereby achieving niche differentiation [97]. Concurrently, niche competition drives the dynamic equilibrium of microbial communities, where certain species may dominate due to their strong adaptability and efficient resource utilization [98], while others may adjust their ecological strategies or seek new niches to cope with competitive pressures. This dynamic balance not only maintains the diversity of microbial communities but also equips them with the ability to respond to environmental changes.” (line 706-721)
“It is noteworthy that the co-occurrence networks of soil bacteria and fungi in the YC region exclusively comprise positive relationships (Fig. S4), a unique phenomenon closely linked to niche differentiation. The specific latitude and longitude of the region, along with its distinctive climatic conditions [107], soil characteristics, and biogeographic patterns [108,109], not only provide abundant nutrients and ideal growth environments for microorganisms but also facilitate niche differentiation among them. The low pH and high levels of total nitrogen (TN), total phosphorus (TP), electrical conductivity (EC), and soil organic carbon (SOC) in YC soils enable different types of microorganisms to select and occupy the most suitable niches based on their environmental adaptability, thereby reducing interspecies competition and promoting community stability and harmony [7]. Furthermore, the unique biogeographic pattern of high latitude and longitude regions offers a relatively stable ecological environment conducive to the coexistence of specific bacteria and fungi, and niche differentiation further minimizes direct competition among them, thus maintaining this positive relationship. Niche differentiation not only enhances microorganisms' resource utilization efficiency but also strengthens community stability and diversity, serving as one of the crucial factors contributing to the positive correlations observed in the soil microbial co-occurrence networks of the YC region.” (line 751-768)
“Soil pH, a crucial factor influencing microbial growth, dictates the adaptability of different bacteria. Among the identified members of soil bacterial communities, some genera, exemplified by Terrimonas and Sphingomonas (Figure 7), exhibit significant positive correlations with soil pH. Terrimonas is adapted to higher pH environments and demonstrates enhanced metabolic activity and growth rates within specific pH ranges [120]. Sphingomonas, on the other hand, engages in complex interactions with other microbial communities in specific pH environments, encompassing symbiosis and competition, which collectively facilitate its survival [121]. Notably, Sphingomonas may collaborate with nitrogen-fixing bacteria that convert atmospheric nitrogen into usable ammonia, providing a valuable nitrogen source for Sphingomonas [122]. In return, Sphingomonas potentially secretes growth factors, vitamins, or other beneficial substances that strengthen this mutualistic relationship [123]. In terms of competition, Sphingomonas' exceptional adaptability to specific pH environments enables it to more efficiently utilize soil resources [124]. This efficient resource utilization not only grants it an advantage in resource competition but also may further inhibit the growth of other competing microorganisms by producing antimicrobial compounds with enhanced activity under specific pH conditions. This competitive advantage ensures Sphingomonas' dominance in soil microbial communities, potentially explaining its higher relative abundance in MIS samples with higher soil pH compared to others (Figure 4a, Table S1). Niche competition, as a significant driver of changes in microbial community structure, not only shapes the success of Sphingomonas but also influences the stability and diversity of the entire soil ecosystem.
In the soil fungal community, the strong negative correlation observed between Tomentella and Pseudosperma with soil pH (Figure 7) highlights an intriguing facet of niche competition. Tomentella produces soil enzymes that exhibit higher activity in acidic environments, which not only promotes its metabolism and growth but also enhances the efficiency of related metabolic pathways [125]. Additionally, acidic conditions may increase the availability of certain nutrients, further aiding the survival and proliferation of Tomentella. This adaptation to acidic soils likely explains their higher relative abundance in YC soils, which have a relatively lower pH. Similarly, the strong negative correlation between Pseudosperma and soil pH can be attributed to the impact of acidic environments on membrane stability [126], with Pseudosperma demonstrating adaptability by maintaining the stability and function of its cell membranes, allowing for normal metabolic activities, ion balance, and preserving the integrity of ion channels and transporters. Conversely, environments with lower acidity or alkalinity can induce conformational changes in membrane lipids and proteins, disrupting membrane fluidity and permeability. Given Pseudosperma's unsuitability for non-acidic environments, this accounts for its high relative abundance in AC soils. These findings emphasize the importance of niche competition in shaping the composition of soil fungal communities, with Tomentella and Pseudosperma securing ecological niches through adaptation to specific pH conditions, providing them with a competitive advantage over other fungi less suited to these environments. This ecological differentiation not only promotes their own survival and growth but also contributes to the overall stability and diversity of the soil ecosystem.” (line 812-855)
“The limited availability of nutrients, water, and organic matter in soil leads to the formation of unique ecological niches among different microbial species, which vary in their efficiency and modes of resource utilization [95]. As environmental conditions such as soil pH and organic matter content change with longitude, the ecological niches of soil fungi and bacteria differentiate accordingly [83]. According to the competitive exclusion principle in niche theory, microorganisms with similar niches compete fiercely for limited resources, potentially leading to the elimination or reduction of some species. To avoid such competitive exclusion, soil microorganisms further differentiate their niches to reduce competitive pressure [98]. The variation in the Chao1 index of soil fungi with longitude reflects this shift in fungal diversity driven by niche differentiation and competition [139]. In resource-rich environments, multiple fungal species may coexist and compete with each other, whereas in resource-poor environments, fungal species diversity decreases, and competitive pressure diminishes [140]. The greater the overlap in niches among fungi, the more intense the competition; conversely, niche differentiation alleviates competition, favoring the coexistence of multiple fungal species [141]. Thus, changes in soil fungal diversity are the joint outcome of soil environmental heterogeneity, microbial niche differentiation, and competitive exclusion.” (line 892-908)
We were deeply grateful for your insights, which undoubtedly improved the quality of our work. Should you have had any further suggestions or comments, we would have eagerly awaited the opportunity to refine our paper further based on your expertise. Thank you once again for your invaluable guidance.
Q5: The co-occurrence network analysis effectively illustrates microbial interactions, but including a comparison with null models or validation steps would help confirm that observed associations are non-random. Incorporating these, as recommended in similar studies, would improve the robustness of the network analysis and provide stronger support for interpreting microbial relationships.
Response 5:
Dear reviewer,
We sincerely appreciate your valuable suggestions on our co-occurrence network analysis work. We deeply felt that your comments were crucial for enhancing the rigor and depth of our research. I am pleased to inform you that we have made corresponding improvements to our study, particularly by adding significant supplements and clarifications to the Materials and Methods section. These additions not only enriched our methodology but also provided a more solid statistical foundation and biological interpretation for deciphering the complex relationships among microorganisms. We believe that, through these improvements, our research now offers stronger support for understanding the structure and function of microbial ecosystems. The specific details are as follows:
“The βNTI values were calculated with the help of “iCAMP”, “ggpubr”, “NST”, “picante” and “ape” packages in the R software (version 4.4.1) [50]. The βNTI values were determined by contrasting the actual community structure with the predicted theoretical structure based on a null model. If the βNTI value falls within the range of -2 to 2, stochastic processes prevail; when the value of βNTI is less than -2 or greater than 2, deterministic processes are prevalent. Under the established constraints, the community data is randomized by reallocating species to different samples or adjusting the interspecies relationships while preserving certain ecological characteristics, such as species richness. To obtain a stable null distribution, the randomization process is typically repeated multiple times (e.g., 1000 times or more) [51]. After each randomization, corresponding community structure indices (e.g., βNTI, RCbray, etc.) are calculated [52]. By constructing the community assembly null model, ecologists can compare the observed community structure indices with the null distribution to identify any discrepancies.” (line 241-254)
Once again, thank you for your invaluable feedback, which has had a positive and profound impact on our study. We look forward to the opportunity to further exchange research findings with you in the future.
We sincerely appreciate your valuable time and expertise in reviewing our manuscript. Your insightful comments and suggestions have been incredibly helpful in improving the quality of our work. The issues you raised are highly professional and objective, and they are of great importance to us. In the process of contemplating, addressing, and responding to the points you highlighted, we have learned a lot and gained considerable insight. This has also been immensely beneficial for our future work endeavors. We hope that the revised manuscript now meets your expectations and the standards of the journal. Thank you once again for your invaluable contribution to our research. We look forward to your further feedback. Taking this opportunity, I, on behalf of my co-authors and myself, would like to express our most sincere blessings to you. May your days be filled with joy, your work with satisfaction, and your life with fulfillment!
Reviewer 3 Report
Comments and Suggestions for Authors
The manuscript entitled "Geographic distribution pattern determines soil microbial community assembly process in Acanthopanax senticosus rhizosphere soil" present a good research on microbial potential assemblages within the rhizosphere community, in relation to a specific plant.
Some suggestions to the current form of the manuscript can be done, in order to improve the presentation of the results and the most important findings for the field of rhizosphere microbiome.
Keywords - try to replace the keywords that are also in the title with similar ones. This will increase the number of links that the manuscript can offer.
Introduction section
The sentence "Soil microbial communities were identified using high-throughput amplicon sequences generated on an Illumina MiSeq platform" belongs to the Materials and Methods section. If you consider that Illumina MiSeq platform as a potential tool to better explore the microbiome changes, please modify the sentence. Something like: An objective was to test if Illumina MiSeq platform is suitable for the detection of community assemblage within different type of rhizosphere. Or a similar formulation.
The introduction present the importance of the study, it has a clear aim and hypotheses presented in clear sentences.
Materials and Methods section - this section describes in detail all the research techniques used by the authors. An interesting presentation is the generous Data analysis section (Bioinformatic and Statistical Analysis) which makes the read of results/discussion, presented in the manuscript, easier to understand and follow. The replication of the experiment is assured by the explanations from this section.
Results section - This section is well detailed, with multiple tables and figures explained by the authors. All the results are explored and the diversity of statistics presented makes the manuscript clear and well presented.
The discussion completes the exploration of the results, by pointing the trends observed by the authors, the mechanism that sustain the assemblage of microbial specific communities. Connections are done with international references.
Conclusion section - suggestions for this section
Reorganize the sentences from this section, to point your results and add some of the main findings. The text from lines 1022-1033 can be the base for the new conclusion section. The rest of the text should be replaced with findings from this study. The future research should be presented in the Discussion section.
Overall, it is and interesting and well written manuscript.
Author Response
Response to Reviewer 2 Comments
Dear Editor,
Dear Reviewer 2,
Thank you very much for giving us the opportunity to revise our manuscript entitled “Geographic distribution pattern determines soil microbial community assembly process in Acanthopanax senticosus rhizosphere soil (microorganisms-3314614)”. Thank you very much for your valuable comments on our MS. According to your comments and suggestions, we carefully revised throughout our MS. Below you will find our point-by-point response.
Comments:
Comments and Suggestions for Authors:
Keywords
Q1: Keywords - try to replace the keywords that are also in the title with similar ones. This will increase the number of links that the manuscript can offer.
Response:
Dear reviewer,
Thank you for your insightful comment. We have carefully reviewed the manuscript and replace the keywords that are also in the title with similar ones. The specific modifications are as follows:
“Acanthopanax senticosus; rhizosphere microbiome; geographic factors; environmental variables” (line 35-36)
Introduction
Q2: The sentence "Soil microbial communities were identified using high-throughput amplicon sequences generated on an Illumina MiSeq platform" belongs to the Materials and Methods section. If you consider that Illumina MiSeq platform as a potential tool to better explore the microbiome changes, please modify the sentence. Something like: An objective was to test if Illumina MiSeq platform is suitable for the detection of community assemblage within different type of rhizosphere. Or a similar formulation.
Response:
Dear reviewer,
Thank you for your insightful comment. We have revised the manuscript based on your feedback. The specific modifications are as follows:
“An objective of this study was to test the suitability of the Illumina Miseq platform for detecting community assemblage within different types of rhizosphere soils.” (line 97-99)
Conclusion
Q3: Reorganize the sentences from this section, to point your results and add some of the main findings. The text from lines 1022-1033 can be the base for the new conclusion section. The rest of the text should be replaced with findings from this study. The future research should be presented in the Discussion section.
Response:
Dear reviewer,
Thank you again for your valuable comments on our manuscript. We have revised the section to highlight the key findings of our study. The text from lines 1022-1033 has been retained as the foundation for the new conclusion. Additionally, we have moved the discussion of future research directions to the Discussion section, as recommended. The specific modifications are as follows:
“These findings contribute to a deeper understanding of the geographic patterns of soil microbial communities associated with cultivated A.senticosus plants.” (line 1101-1102)
We sincerely appreciate your valuable time and expertise in reviewing our manuscript. Your insightful comments and suggestions have been incredibly helpful in improving the quality of our work. The issues you raised are highly professional and objective, and they are of great importance to us. In the process of contemplating, addressing, and responding to the points you highlighted, we have learned a lot and gained considerable insight. This has also been immensely beneficial for our future work endeavors. We hope that the revised manuscript now meets your expectations and the standards of the journal. Thank you once again for your invaluable contribution to our research. We look forward to your further feedback. Taking this opportunity, I, on behalf of my co-authors and myself, would like to express our most sincere blessings to you. May your days be filled with joy, your work with satisfaction, and your life with fulfillment!
Best Regards
Yours Sincerely,
Dr. Xin Sui
Engineering Research Center of Agricultural Microbiology Technology, Ministry of Education & Heilongjiang Provincial Key Laboratory of Ecological Restoration and Resource Utilization for Cold Region & Key Laboratory of Microbiology, College of Heilongjiang Province & School of Life Sciences, Heilongjiang University, Harbin 150080, China. xinsui_cool@126.com (X. Sui).
Round 2
Reviewer 1 Report
Comments and Suggestions for Authors
No comments.